# Introduction to some of the simplest topological phases of matter

## Frank Schindler[1]

[1] Blackett Laboratory, Imperial College London, London SW7 2AZ, United Kingdom

frankschindler.github.io
Please reach out if you find errors! :)

These lecture notes explain the classification of some simple fermionic topological phases of matter in a pedestrian manner, with an aim to be maximally pedagogical = doing things in excruciating detail. We focus on a many-body perspective, even if many of the models we work with are non-interacting. We start out with symmetry protected topological (SPT) phases of free fermions that are protected by U(1) symmetry = topological insulators. We then look at fermion topological phases that don't even need a symmetry = topological superconductors, and explain how their classification changes in presence of spinless time-reversal symmetry. We close by perturbatively checking which of the 1D topological phases we had found are stable to interactions.

## 1    Introduction

Warning: the language in this section is a bit imprecise and only meant to give a first impression. I promise that everything will become clear with the explicit examples discussed later.

### 1.1    What is a SPT phase?

By a quantum *phase of matter*, we mean a smoothly connected family of ground states of quantum many-body Hamiltonians at zero temperature.

What do we mean by smooth? Generalising from the notion of *n*-th order phase transitions in thermodynamics, we mean that *all* expectation values computed in the ground state remain continuous, including the energy. This behaviour is guaranteed when the Hamiltonian is a smooth function, meaning all its derivatives are continuous, *and* the ground state is gapped. Note that the converse is not necessarily true – gapless phases of matter exist as well.

The ground states associated with symmetry-protected topological (SPT) phases satisfy two defining properties:

(1)  On closed manifolds[1], SPT states are the *unique* gapped ground states of local Hamiltonians $H$ that preserve a given symmetry group $\mathcal{G}$, so that $[g, H] = 0$ for all group elements $g \in \mathcal{G}$. Note in particular that the condition of uniqueness forbids spontaneous symmetry breaking.

[1] For our purposes, closed manifold is just a fancy way of saying 'periodic boundary conditions', meaning a $d$-dimensional torus.

(2)  SPT states cannot be smoothly deformed to a *trivial reference state* without breaking the symmetry.

We'd usually like to define the trivial reference state as a product state *in position space* = a state that does not exhibit entanglement between different locations. For some symmetry classes, this choice is not unique. This makes the definition of SPTs a matter of convention. We should then really think of SPT order as a relative distinction that guarantees certain families of ground states are not smoothly connected to each other, rather than as an absolute property.

Note that SPT phases *can* be smoothly deformed to a trivial reference state without a gap closing = without a phase transition, as long as their protecting symmetry is broken somewhere along the path. As the trivial reference state has no entanglement by definition, SPT phases are therefore also called short-range entangled phases, to distinguish them from long-range entangled *topologically ordered* states that do not require a symmetry for their stability.

SPT phases can be defined for interacting bosons, for instance in the context of spin chains (keywords: Haldane spin chain, AKLT

chain). In the present notes, however, we focus on fermionic systems that describe quantum materials built from electrons and ionized atoms. One practical aspect of working with fermions is that they do not require interactions to give rise to nontrivial ground states. The basic reason for this is that the Pauli exclusion principle prevents fermions from occupying the same quantum state, so that the physical properties of many-body fermionic states can differ significantly from those of any single-particle state. We will start our exploration of SPT phases with non-interacting = free fermion systems, and then upgrade to interactions later on.

## 1.2  Notation

Before we start out, let's introduce our notation. We will study quantum systems built from $N$ fermionic modes. For example, when the system is a one-dimensional (1D) chain, then $N \propto L$, where $L$ is the length of the chain.

The fermions are annihilated by the operators $c_i$, $i = 1 \ldots N$, and created by their Hermitian conjugates $c_i^\dagger$, so that the canonical anti-commutation relations hold:

$$\{c_i, c_j\} = 0, \quad \{c_i, c_j^\dagger\} = \delta_{ij}, \quad \{c_i^\dagger, c_j^\dagger\} = 0. \tag{1.1}$$

Here $\{X, Y\} = XY + YX$ is the anticommutator and $\delta_{ij}$ is the Kronecker delta that is equal to 1 when $i = j$ and zero otherwise.

These anti-commutation relations imply the Pauli exclusion principle: there is no state that contains more than one fermion in the same mode, $(c_i^\dagger)^2 = 0$.

The fermionic vacuum state $|0\rangle$ is defined by the fact that it is annihilated by *all* of the $c_i$, *i.e.*, it does not contain any fermions:

$$c_i |0\rangle = 0, \quad \langle 0|0\rangle = 1. \tag{1.2}$$

Note that the second condition (normalisation) is important for this to describe a physical state, as otherwise $|0\rangle = 0$ would be a solution.

In the following, we will often colloquially refer to the fermions as *electrons* or *particles*.

## 2    Free fermion SPTs with charge conservation symmetry

*Free* fermions are non-interacting fermions – the only way that such particles can influence one another is the Pauli exclusion principle. In general, they are described by a Hamiltonian operator of the form

$$H = \sum_{ij} \left( \mathcal{H}_{ij} c_i^\dagger c_j + \Delta_{ij} c_i^\dagger c_j^\dagger + \Delta_{ij}^* c_j c_i \right), \tag{2.1}$$

where $\mathcal{H}_{ij} = \mathcal{H}_{ji}^* \in \mathbb{C}$ to ensure Hermiticity $H^\dagger = H$, and $\Delta_{ij} \in \mathbb{C}$. Physically, $\mathcal{H}_{ij}$ usually corresponds to fermions hopping between different atomic sites and orbitals of a crystal, and $\Delta_{ij}$ incorporates their superconducting pairing within a mean-field description. Note that the many-body Hamiltonian $H$ is a $2^N \times 2^N$ matrix – each of the $N$ fermionic modes can be empty or occupied – while the hopping matrix $\mathcal{H}$ (sometimes also called the single-particle Hamiltonian), and the pairing matrix $\Delta$, are $N \times N$ matrices.

Imposing locality, linear terms in the fermion operators (*e.g.*, $c_i + c_i^\dagger$) are physically impossible in $H$. This is because operators containing an odd number of fermions anti-commute with each other irrespective of their distance, whereas locality requires that terms in the Hamiltonian that act on distant locations commute. Moreover, our restriction to free fermions implies that $H$ does not contain inter-action terms such as $\hat{n}_i \hat{n}_j$, where $\hat{n}_i = c_i^\dagger c_i$ is the number operator. (For example, such terms would show up in the operator for the Coulomb interaction.) For this reason, free fermion Hamiltonians are also sometimes called *quadratic* fermion Hamiltonians.

**Charge conservation:** Eq. (2.1) already looks a bit complicated so lets start by looking at the simpler case of free fermions with *charge conservation symmetry* – also often called U(1) symmetry: defining the total particle number operator by[1]

$$\hat{N} = \sum_i \hat{n}_i, \quad \hat{n}_i = c_i^\dagger c_i, \tag{2.2}$$

the requirement of charge conservation symmetry translates to $[\hat{N}, H] = 0$ and therefore we must have $\Delta = 0$. Since a general unitary U(1) symmetry transformation is generated by $\hat{N}$ for some $\theta \in \mathbb{R}$,

$$U(\theta) = e^{i\theta \hat{N}} \in U(1), \tag{2.3}$$

the vanishing commutator $[\hat{N}, H] = 0$ implies that in fact *all* elements of the U(1) symmetry group commute with the Hamiltonian, satisfying condition (1) in Sec. 1.

Applying the U(1) symmetry condition to Eq. (2.1), we end up with the most general charge-conserving free fermion Hamiltonian

$$H = \sum_{ij} \mathcal{H}_{ij} c_i^\dagger c_j. \tag{2.4}$$

[1] Note that the particle number operator $\hat{N}$ is *not* the same as the number of fermionic modes $N$. (Apologies for overcrowding the notation!) Due to the Pauli exclusion principle however, in any state of the Hilbert space we must have $\langle \hat{N} \rangle \leq N$.

We now want to find the ground state of this Hamiltonian to study the possible free fermion SPT phases in presence of U(1) symmetry. First, define the eigenvectors $\phi^\alpha$ of the $N \times N$ matrix $\mathcal{H}$ by

$$\sum_j \mathcal{H}_{ij}\phi_j^\alpha = \epsilon_\alpha \phi_i^\alpha. \tag{2.5}$$

These lend themselves to the definition of new *quasiparticles*

$$d_\alpha^\dagger = \sum_j \phi_j^\alpha c_j^\dagger. \tag{2.6}$$

Using that the $\phi^\alpha$ are orthonormal, which follows from the fact that they are eigenvectors of a *Hermitian* matrix, it is easy to see that the quasiparticles satisfy the canonical anti-commutation relations

$$\{d_\alpha, d_\beta\} = 0, \quad \{d_\alpha, d_\beta^\dagger\} = \delta_{\alpha\beta}, \quad \{d_\alpha^\dagger, d_\beta^\dagger\} = 0. \tag{2.7}$$

A bit of algebra shows that the Hamiltonian becomes diagonal in the quasiparticles

$$H = \sum_\alpha \epsilon_\alpha d_\alpha^\dagger d_\alpha. \tag{2.8}$$

The eigenstates of $H$ are therefore product states of the form[2]

$$|\boldsymbol{n}\rangle = (d_1^\dagger)^{n_1}(d_2^\dagger)^{n_2}\cdots(d_N^\dagger)^{n_N}|0\rangle, \tag{2.9}$$

where $\boldsymbol{n}$ is a vector of occupation numbers $n_\alpha = 0,1$ and higher occupations are impossible due to Pauli exclusion. As expected, there are $2^N$ different states depending on the different choices of $\boldsymbol{n}$. These states have the energy

$$H|\boldsymbol{n}\rangle = \left(\sum_\alpha \epsilon_\alpha n_\alpha\right)|\boldsymbol{n}\rangle \equiv E_{\boldsymbol{n}}|\boldsymbol{n}\rangle, \tag{2.10}$$

which can be easily seen by noting that $d_\alpha^\dagger d_\alpha = \hat{n}_\alpha$ in Eq. (2.8) is nothing but the quasiparticle number operator for the mode $\alpha$.

Homework: show that the number operator $\hat{N}$ assumes the same form in the original fermions and the quasiparticles:

$$\hat{N} = \sum_i c_i^\dagger c_i = \sum_\alpha d_\alpha^\dagger d_\alpha. \tag{2.11}$$

Therefore, the states $|\boldsymbol{n}\rangle$ have a fixed particle number:

$$\hat{N}|\boldsymbol{n}\rangle = \left(\sum_\alpha n_\alpha\right)|\boldsymbol{n}\rangle. \tag{2.12}$$

The *ground* state $|\text{GS}\rangle$ of $H$ is the state of minimal $E_{\boldsymbol{n}}$. If all $\epsilon_\alpha > 0$, this is the vacuum state, $|\text{GS}\rangle = |0\rangle$. If some $\epsilon_\alpha < 0$ are negative, the ground state has those quasiparticles occupied:

$$|\text{GS}\rangle = \prod_\alpha^{\epsilon_\alpha < 0} d_\alpha^\dagger |0\rangle. \tag{2.13}$$

[2] Such states are also called Slater determinants: their associated many-particle wavefunction in real space takes the form of a determinant, which follows from fermionic statistics.

Note that this ground state is not necessarily topologically trivial in the SPT sense (see Sec. 1.1), even though it is a product state: the product is over the abstract quasiparticles that need not be localised in position space.

If some quasiparticle energies are zero, $\epsilon_\gamma = 0$, then we have a ground state degeneracy where states such as $d_\gamma^\dagger |\text{GS}\rangle$ [where $|\text{GS}\rangle$ remains defined by Eq. (2.13)] minimise the energy $E_n$ equally well. In this case, we really have to think about the full set of ground states together. By definition (Sec. 1.1), this situation does not arise for SPT phases on closed manifolds.

In presence of charge conservation symmetry and absent degeneracy, the *energy gap $\delta$* above the ground state is defined as the energy difference between the ground state and the next lowest eigenstate of $H$ that has the same number of particles. Sorting the single-particle energies so that $\epsilon_1 \leq \epsilon_2 \leq \cdots \leq \epsilon_N$, and assuming $m > 0$ particles are occupied in $|\text{GS}\rangle$ so that $\epsilon_m < 0$ and $\epsilon_{m+1} > 0$, the gap is given by

$$\delta = \epsilon_{m+1} - \epsilon_m. \tag{2.14}$$

We are now ready to work out the possible free fermion SPT phase protected by U(1) symmetry in different dimensions of space.

## 2.1   0D

It is instructive to look at zero spatial dimensions (0D) first – for example, a quantum dot described by a Hamiltonian of the form of Eq. (2.4). In this scenario, the indices $i, j$ range over different atomic sites, orbitals, and spin degrees of freedom that *electrons* can occupy. (We do not include the *atomic ions* in our quantum mechanical treatment, because they are much heavier than the electrons and can usually be assumed to be frozen in place[3].)

Due to the smallness of the quantum dot (a 0D system is not extensive in any direction of space), there is no notion of locality. We can still define "SPT" phases in 0D by fixing the trivial reference state to be the fermionic vacuum $|0\rangle$. The other states of the Hilbert space, which have a non-zero particle number, are then clearly distinct from the vacuum: they cannot be smoothly connected to it as long as charge conservation symmetry is preserved. To see this, note that any Hamiltonian $H$ of the form of Eq. (2.4) is block diagonal in particle number, meaning that we can choose its eigenstates to also be eigenstates of $\hat{N}$. Since the eigenvalues of $\hat{N}$ are integers, they cannot change smoothly. More generally, we cannot smoothly deform *any* two ground states into one another as long as their particle number is different.

How about two ground states that have the same particle num-

[3] This is called the Born-Oppenheimer approximation.

ber? To study this case, let us consider a general hopping matrix $\mathcal{H}$. This matrix fully determines the U(1)-preserving Hamiltonian $H$ in Eq. (2.4). We have seen before that the number of negative eigenvalues $m$ of $\mathcal{H}$ is equal to the number of fermions in the ground state of $H$. Without closing the gap or breaking U(1) symmetry, we can smoothly deform this to a situation where $\mathcal{H}$ has $m$ eigenvalues $-1$ and $l = (N - m)$ eigenvalues $+1$. The most general such matrix $\mathcal{H}$ takes the form

$$\mathcal{H} = U \begin{pmatrix} -\mathbb{1}_{m \times m} & 0 \\ 0 & \mathbb{1}_{l \times l} \end{pmatrix} U^\dagger, \tag{2.15}$$

where $U$ is the $N \times N$ unitary matrix that diagonalises $\mathcal{H}$ [its columns are the eigenvectors $\phi^\alpha$ in Eq. (2.5), $U_{i\alpha} = \phi_i^\alpha$].

One might think that there is a one-to-one correspondence between $\mathcal{H}$ and $U$, but this is incorrect – there is a $U(m) \times U(l)$ gauge symmetry that mixes eigenstates within each of the eigenvalue $-1$ and $+1$ subspaces, without changing $\mathcal{H}$. In particular, we can always rewrite

$$\begin{pmatrix} -\mathbb{1}_{m \times m} & 0 \\ 0 & \mathbb{1}_{l \times l} \end{pmatrix} = \begin{pmatrix} U_{m \times m} & 0 \\ 0 & U_{l \times l} \end{pmatrix} \begin{pmatrix} -\mathbb{1}_{m \times m} & 0 \\ 0 & \mathbb{1}_{l \times l} \end{pmatrix} \begin{pmatrix} U_{m \times m}^\dagger & 0 \\ 0 & U_{l \times l}^\dagger \end{pmatrix}, \tag{2.16}$$

for some arbitrary choice of unitary matrix blocks $U_{m \times m}$ and $U_{l \times l}$. This amounts to no change in $\mathcal{H}$ but a change in $U$:

$$U \to U' = U \begin{pmatrix} U_{m \times m} & 0 \\ 0 & U_{l \times l} \end{pmatrix} \tag{2.17}$$

Correspondingly, the space of all possible matrices $\mathcal{H}$ for which there are $m$ particles in the ground state corresponds to the unitary group U($N$) of $N \times N$ unitary matrices, *modulo* the equivalence relation in Eq. (2.17): all unitary matrices $U$ that differ by this relation should be identified. The remaining *inequivalent* elements of this space are then elements of the complex *Grassmannian manifold*

$$\mathrm{Gr}(m, \mathbb{C}^N) = \mathrm{U}(N) / \left[ \mathrm{U}(m) \times \mathrm{U}(N - m) \right], \tag{2.18}$$

where the symbol "/" denotes the quotient operation of groups. A little thought shows that this manifold is connected – writing $U = e^{iX}$ where $X$ is Hermitian, we can smoothly tune the entries of the matrix $X$ to reach all points of $\mathrm{Gr}(m, \mathbb{C}^N)$, potentially many times over due to the equivalence relation. Therefore, all ground states with the same particle number are smoothly connected to each other.

In conclusion, "SPT" phases of free fermions with U(1) symmetry in 0D have a $\mathbb{Z}$ classification – there is a different phase for every integer. They are differentiated by $\langle \hat{N} \rangle \in \mathbb{Z}$, the number of particles in the ground state.[4]

[4] Technically, particle numbers measured from the vacuum must lie in $\mathbb{Z}^+$, but there is a bijection between this set and $\mathbb{Z}$; the usual convention is to just use $\mathbb{Z}$ here.

## 2.2   1D

Consider next a one-dimensional (1D) chain described by Eq. (2.4), where $i = 1 \ldots N$ now denotes different sites arranged in a 1D lattice: $i = 1$ corresponds to a site at the left end of the lattice, while $i = N$ is located at the right end.

Again, U(1) symmetry implies that 1D ground states with different particle numbers cannot be smoothly connected to one another. Since this distinction is already present in 0D, we do not consider it part of the classification of 1D SPT phases. In fact, any $n$-dimensional system can technically be viewed as a "fat" $(n - 1)$-dimensional system. Therefore, the classification of SPT phases in a given dimension also applies to all higher dimensions, albeit in a rather uninteresting way. The actually interesting question is if higher dimensions allow for new SPT phases, enabled by locality, that are impossible in lower dimensions.[5]

What do we mean by locality? Roughly we mean that $H$ in Eq. (2.4) does not move fermions between locations that are far apart in space. Formally, one usually requires that the hopping matrix elements $\mathcal{H}_{ij}$ decay exponentially with the separation between $i$ and $j$, or that they are strictly zero beyond a maximal radius.

***Bloch Hamiltonian and band structure***   For simplicity, let us now assume translational symmetry, we will relax this assumption later. The hopping matrix $\mathcal{H}$ then becomes block-diagonal in *crystal momentum* $k$, allowing us to define a *Bloch Hamiltonian* $\mathcal{H}(k)$. The requirement of locality translates to the statement that $\mathcal{H}(k)$ varies smoothly with $k$.[6]

Well, this was a bit imprecise, so let's go through it in detail. We don't just want *full* translational symmetry – meaning a symmetry that translates $i \to i + 1$, as this might be too restrictive. Consider for example a spinful system where each lattice site has two possible spin degrees of freedom. In this case, translations between different sites would transform $i \to i + 2$, because a translation does not flip the spin. The general way to handle this is to introduce a *unit cell* that contains all the things that are not translates of one another (spins, orbital degrees of freedom, sublattices), then the full lattice is generated by translating the unit cell.

To achieve this mathematically, we first write the index $i$ as a composite index $i = (R, a)$, where $R = 1 \ldots L$ is the unit cell coordinate (we assume an integer lattice spacing without loss of generality), and $a = 1 \ldots M$ ranges over the $M$ different degrees of freedom in each unit cell. We then have $LM = N$. Note that only the index $R$ has translational symmetry, at least as long as we implement periodic boundary conditions (PBC) by setting $c^\dagger_{L+1,a} \equiv c^\dagger_{1,a}$. We can then

[5] Since locality imposes *additional* constraints on the ground state, it cannot trivialise the lower-dimensional classification: consider two ground states that realise different 0D SPT phases. By definition, *all* paths connecting the two states are discontinuous somewhere, or break the symmetry. Imposing locality then just restricts us to a subset of these paths.

[6] The technical statement is that $\mathcal{H}(k)$ is an analytic function of $k$, meaning it has a converging Taylor series expansion.

define the momentum space creation operators by

$$c_{k,a}^\dagger = \frac{1}{\sqrt{L}} \sum_R e^{ikR} c_{R,a}^\dagger \quad \to \quad c_{R,a}^\dagger = \frac{1}{\sqrt{L}} \sum_k e^{-ikR} c_{k,a}^\dagger, \tag{2.19}$$

and you can verify that these still satisfy the usual anti-commutation relations. Importantly, since our lattice has periodicity $L$, the momentum sum only runs over the allowed values

$$k \in \frac{2\pi}{L}\mathbb{Z} \quad \to \quad c_{R+L,a}^\dagger = c_{R,a}^\dagger. \tag{2.20}$$

Conversely, since our lattice sites have integer spacing by assumption, $R \in \mathbb{Z}$, a momentum shift $k \to k + 2\pi$ clearly does not affect $c_{k,a}^\dagger$. In the thermodynamic limit $L \to \infty$, the inequivalent crystal momenta $k \in [0, 2\pi)$ then live in the 1D *Brillouin zone* that is topologically equivalent to the unit circle $S^1$.

Translational symmetry implies that

$$\mathcal{H}_{(R,a),(R',b)} = \mathcal{H}_{(R-R',a),(0,b)} \tag{2.21}$$

so that we find:

$$\begin{aligned}
H &= \sum_{R,R',a,b} \mathcal{H}_{(R,a),(R',b)} c_{R,a}^\dagger c_{R',b} \\
&= \frac{1}{L} \sum_{R,R',a,b} \sum_{k,q} e^{-i(kR-qR')} \mathcal{H}_{(R,a),(R',b)} c_{k,a}^\dagger c_{q,b} \\
&= \frac{1}{L} \sum_{R,R',a,b} \sum_{k,q} e^{-i(k(R+R')-qR')} \mathcal{H}_{(R+R',a),(R',b)} c_{k,a}^\dagger c_{q,b} \\
&= \frac{1}{L} \sum_{R,R',a,b} \sum_{k,q} e^{-i(kR+(k-q)R')} \mathcal{H}_{(R,a),(0,b)} c_{k,a}^\dagger c_{q,b} \\
&= \sum_{R,a,b} \sum_k e^{-ikR} \mathcal{H}_{(R,a),(0,b)} c_{k,a}^\dagger c_{k,b} \\
&= \sum_{k,a,b} \underbrace{\left( \sum_R e^{-ikR} \mathcal{H}_{(R,a),(0,b)} \right)}_{\mathcal{H}(k)_{ab}} c_{k,a}^\dagger c_{k,b}.
\end{aligned} \tag{2.22}$$

Here we have used PBC to shift summation variables at will, and we have also used the identity

$$\sum_R e^{i(k-q)R} = L\delta_{k,q}. \tag{2.23}$$

From this definition of $\mathcal{H}(k)$, we see that longer range hoppings, meaning a non-zero $\mathcal{H}_{(R,a),(0,b)}$ for larger values of $|R|$, correspond to more rapid oscillations $e^{-ikR}$ in momentum. Locality guarantees that the amplitude of these oscillations decays fast enough to result in a $\mathcal{H}(k)$ that's a smooth function of $k$. This condition will turn out to be important in stabilising 1D SPT phases.

Diagonalising the $M \times M$ *Bloch Hamiltonian* $\mathcal{H}(k)$ at every momentum $k$ returns a band structure of single-particle energies $\epsilon_\mu(k)$, $\mu = 1 \ldots M$, that continuously disperse with momentum $k$ in the Brillouin zone. The many-body ground state of $H$, as defined in Eq. (2.13), is then formed by occupying all single-particle eigenstates where $\epsilon_\mu(k) < 0$ is negative, at all momenta $k$. This ground state is gapped as long as no band crosses zero energy.

Since 'zero energy' is an arbitrary concept, we should really define the ground state simply as occupying *all* states in, say, the $m$ lowest energy bands, $\mu = 1 \ldots m$. As long as these bands are separated from the rest of the band structure by a gap at every momentum $k$, we can smoothly deform this band structure to agree also with the previous definition.

*Dirac theory of gap closing*    To classify the possible 1D SPT phases, the basic idea is to start from a generic phase transition = a gap closing, and determine how many distinct (= not smoothly connected) ground states we can gap out. A little thought shows that without fine tuning or additional symmetries, such band gap closings in the spectrum of $\mathcal{H}(k)$ generically happen at a *point* $k = k_0$ in momentum space – requiring a gap closing on a whole line would impose additional and unnecessary constraints on the band structure. Moreover, without loss of generality, we can assume that the gap closing happens at zero energy[7]. Restricting to the $M_0 \leq M$ bands that participate in the gap closing, we can expand the band structure around the gap closing momentum $k_0$ to first order in $q = k - k_0$ to arrive at the continuum model[8]

$$h(q) = \gamma_1 q + \mathcal{O}(q^2), \qquad (2.24)$$

where $\gamma_1$ is a $M_0 \times M_0$ matrix[9], and there is no constant term $\gamma_0$ because we assumed that the gap closes at zero energy when $q = 0$.

The spectrum of $h(q)$ is given by the eigenvalues of the matrix $\gamma_1$, times $q$. To have a gap closing *only* at $q = 0$, all eigenvalues of $\gamma_1$ should therefore be non-zero. Moreover, the sign of each eigenvalue of $\gamma_1$ determines whether the energy $\epsilon(q)$ of the corresponding eigenmode increases or decreases with momentum $q$. We say the corresponding mode is a right-mover (positive sign) or a left-mover (negative sign), because the semiclassical group velocity of a Bloch wave is defined by $v = \partial \epsilon(q) / \partial q$.

We know that the full band structure of $\mathcal{H}(k)$ must be periodic in the Brillouin zone. At the same time, by assumption, $h(q)$ contains all $M_0$ bands that cross the band gap[10]. Therefore, we must have an equal number $M_0/2$ of right- and left-movers, implying that $M_0$ is always even. Working in the eigenbasis of $\gamma_1$, it follows that we can

[7] By abuse of notation we here speak of single-particle energies, which are the eigenvalues of $\mathcal{H}(k)$; we do *not* mean the energy of the corresponding many-body ground state.

[8] We call this a continuum model because this Hamiltonian does not know about the crystal lattice anymore – the momentum $q$ is not constrained to the Brillouin zone and instead lives on the real line. Physically, this corresponds to the statement that very small momentum differences $q$ correspond to wavelengths that are much larger than the lattice spacing, and therefore insensitive to lattice details.

[9] Formally, $\gamma_1$ is obtained in first-order perturbation theory from the matrix elements of the operator $\partial \mathcal{H}(k)/\partial k|_{k=k_0}$, expressed in the basis formed by the $M_0$ zero-energy eigenstates of $\mathcal{H}(k_0)$.

[10] If a gap closing happens at multiple points in the Brillouin zone, these can always be brought together to the same point by a smooth deformation.

smoothly deform $\gamma_1$ to have the form

$$\gamma_1 = \sigma_z \otimes \mathbb{1}, \tag{2.25}$$

where $\sigma_z$ is a $2 \times 2$ Pauli matrix, $\otimes$ denotes the Kronecker product, and $\mathbb{1}$ is shorthand for the $(M_0/2) \times (M_0/2)$ identity matrix. Note in particular that this result implies that $(\gamma_1)^2 = 1$.

We'd now like to fully gap the spectrum of $h(q) = \gamma_1 q$, to result in a gapped ground state that is a candidate for a 1D SPT phase. Assuming the perturbed Hamiltonian

$$h(q) = \gamma_1 q + \gamma_2, \tag{2.26}$$

for $\gamma_2$ some $M_0 \times M_0$ matrix, how must we choose $\gamma_2$ to yield a gap? First of all, $\gamma_2$ must clearly have all eigenvalues non-zero to open a gap at $q = 0$. Up to smooth deformations, we can assume that these eigenvalues are $\pm 1$, meaning that $(\gamma_2)^2 = 1$.

Next, we consider the gap condition away from $q = 0$. Denote by $|\phi_\mu^\pm\rangle$ the eigenstates of $\gamma_1$ with eigenvalue $\pm 1$, where $\mu = 1 \ldots M_0/2$. In the unperturbed system, these states disperse with an energy $\epsilon_\mu^\pm(q) = \pm q$ away from the gap closing. A little thought shows that to open a gap at *all* momenta $q$, we need to couple the right-movers and left-movers with each other, instead of just shifting their individual energies. This means that the block off-diagonal matrix elements

$$\langle \phi_\mu^+ | \gamma_2 | \phi_\nu^- \rangle \neq 0 \tag{2.27}$$

should be non-zero. Moreover, we can then always set the block diagonal matrix elements to zero,

$$\langle \phi_\mu^+ | \gamma_2 | \phi_\nu^+ \rangle = \langle \phi_\mu^- | \gamma_2 | \phi_\nu^- \rangle = 0, \tag{2.28}$$

while keeping the gap open. Stating the same in a basis-independent fashion, the condition to open a gap at all momenta is equivalent (at least up to smooth deformations) to requiring that $\gamma_2$ anti-commutes with $\gamma_1$,

$$\{\gamma_1, \gamma_2\} = 0. \tag{2.29}$$

Since we have already seen that $(\gamma_1)^2 = 1 = (\gamma_2)^2$, this means that the $\gamma$ matrices form a Clifford algebra, so that Eq. (2.26) is a condensed matter realisation of the 1D Dirac equation.

*Topology of the space of possible Dirac mass terms*   How many inequivalent choices of $\gamma_2$ are there? Given that $\gamma_1 = \sigma_z \otimes \mathbb{1}$, the most general matrix that satisfies the above requirements for $\gamma_2$ reads

$$\gamma_2 = e^{i(\sigma_0 \otimes A + \sigma_z \otimes B)} \left( \sigma_x \otimes \mathbb{1} \right) e^{-i(\sigma_0 \otimes A + \sigma_z \otimes B)}, \tag{2.30}$$

where $A$ and $B$ are arbitrary Hermitian $(M_0/2) \times (M_0/2)$ matrices. This result requires a bit of thought. First of all, note that the matrix $\sigma_x \otimes \mathbb{1}$ definitely has the desired properties, in that it squares to 1 and anticommutes with $\gamma_1$. Furthermore, the general unitary transformation $e^{i(\sigma_0 \otimes A + \sigma_z \otimes B)}$ commutes with $\gamma_1 = \sigma_z \otimes \mathbb{1}$ and therefore preserves the anti-commutation relation. Finally, instead of $\sigma_x \otimes \mathbb{1}$, why did we not start out with the more general form $\sigma_x \otimes C$, where $C$ is some Hermitian $(M_0/2) \times (M_0/2)$ matrix satisfying $C^2 = 1$? In the eigenbasis of $C$, such a matrix would have the form

$$\tilde{\gamma} = \sigma_x \otimes \begin{pmatrix} \mathbb{1}_{s \times s} & 0 \\ 0 & -\mathbb{1}_{t \times t} \end{pmatrix}, \qquad (2.31)$$

where $s + t = M_0/2$. Note that this matrix has exactly the same set of eigenvalues as our choice above, $\sigma_x \otimes \mathbb{1}$, and so there should be a unitary that relates the two. Indeed, consider the unitary

$$V = e^{i \frac{\pi}{2} \sigma_z \otimes X}, \quad X = \begin{pmatrix} 0_{s \times s} & 0 \\ 0 & \mathbb{1}_{t \times t} \end{pmatrix}, \qquad (2.32)$$

then it is easy to verify $V \tilde{\gamma} V^\dagger = \sigma_x \otimes \mathbb{1}$. Since $V^\dagger$ corresponds to just one particular choice $A = 0$, $B = -\pi X/2$ for the unitary in Eq. (2.30), we can use $\sigma_x \otimes \mathbb{1}$ in that expression without loss of generality.

Since we can tune $A$ and $B$ in Eq. (2.30) smoothly to reach all possible choices for $\gamma_2$, all resulting gapped ground states are smoothly connected to each other. We deduce that U(1) charge conservation symmetry does *not* protect distinct SPT phases in 1D free fermion systems.

Where have we used locality = smoothness of $\mathcal{H}(k)$ to derive this result? First, we have assumed that $\mathcal{H}(k)$ has a well-behaved Taylor expansion in Eq. (2.24). Then, we have used the periodicity and smoothness of the eigenvalue bands of $\mathcal{H}(k)$ in the Brillouin zone to show that $\gamma_1$ has the same number of positive and negative eigenvalues, *i.e.*, Eq. (2.25). Both properties do not hold anymore when $\mathcal{H}(k)$ becomes a discontinuous function of $k$. At this point, everything goes, and we're back to the 0D classification.

***Broken translational symmetry***   What happens when we relax the condition of translational symmetry? We now give a handwavy argument that the classification remains the same – see Kitaev[11] for a more rigorous proof. We can relax translational symmetry step by step by increasing our unit cell: this increases the total number of bands $M$ in the band structure and therefore in general also the number of bands $M_0$ participating in the gap closing. In fact, we can make our unit cell arbitrarily large, which will correspond to an arbitrarily large matrix $h(q)$ in Eq.(2.26). Since our previous arguments

[11] Alexei Kitaev.  Periodic table for topological insulators and superconductors.  *AIP Conference Proceedings*, 1134(1):22–30, 05 2009.  ISSN 0094-243X.  DOI: 10.1063/1.3149495.  URL https://doi.org/10.1063/1.3149495

did not depend on the number of bands, they are not affected by this and the classification remains the same. This argument will break down as soon as the size of the unit cell becomes a finite fraction of the total system size, because then momentum is not a continuous variable anymore. One might ask if there is a SPT phase that can only exist if translational symmetry is broken at such a large scale. We will not pursue this possibility here, as such a phase would need to be inherently non-local to notice this symmetry breaking.

## 2.3   2D

We can immediately generalise our classification method to higher dimensions. In two spatial dimensions (2D), there are now two momenta $q_x$ and $q_y$ and Eq. (2.26) becomes the 2D Dirac equation

$$h(q_x, q_y) = \gamma_1 q_x + \gamma_2 q_y + \gamma_3, \tag{2.33}$$

where the matrices $\gamma_l$, $l = 1 \ldots 3$, again form a Clifford algebra:

$$\{\gamma_l, \gamma_m\} = 2\delta_{mn}. \tag{2.34}$$

Note in particular that $\gamma_1$ and $\gamma_2$ must anti-commute so that, absent $\gamma_3$, there is a gap at all momenta *except* for $q_x = q_y = 0$. Then, to completely gap out the spectrum, $\gamma_3$ must anti-commute with both of these matrices. Furthermore, in each direction of momentum space, the number of left- and right-movers must again be equal, so that we can choose

$$\gamma_1 = \sigma_z \otimes \mathbb{1}, \quad \gamma_2 = \sigma_y \otimes \mathbb{1}. \tag{2.35}$$

What is the most general matrix $\gamma_3$ that satisfies all the required properties, including that it should square to 1? We might naively want to write it as

$$\gamma_3 = \sigma_x \otimes \mathbb{1}, \tag{2.36}$$

where comparing with Eq. (2.30) there is no extra unitary transformation that we need to bracket this in: There is no nontrivial $2 \times 2$ matrix that commutes with *both* $\sigma_z$ and $\sigma_y$, meaning that $\gamma_3$ can only involve $\sigma_x$. However, recall from Eq. (2.31) and the surrounding discussion that we had used the unitary degree of freedom present in Eq. (2.30) to avoid the more general form Eq. (2.31). We do not have this luxury anymore in 2D, and so the correct and most general choice of $\gamma_3$ actually reads

$$\gamma_3 = e^{i\sigma_0 \otimes A} \left[ \sigma_x \otimes \begin{pmatrix} \mathbb{1}_{s \times s} & 0 \\ 0 & -\mathbb{1}_{t \times t} \end{pmatrix} \right] e^{-i\sigma_0 \otimes A}, \tag{2.37}$$

where $s + t = M_0/2$ and $A$ is again some arbitrary $M_0/2 \times M_0/2$ Hermitian matrix. We see that this situation is precisely equivalent to

the 0D case, Eq. (2.15), and therefore conclude that there is a different ground state for every choice of the integer $s$. Since these ground states cannot be smoothly connected to one another, they are all topologically distinct. Moreover, as we can in principle consider an arbitrary number of bands $M_0$ participating in the gap closing, $s$ can take on any (non-negative) integer value.

As a consequence, 2D free fermion SPT phases with U(1) symmetry are $\mathbb{Z}$ classified.[12] They are differentiated by a topological invariant called the *Chern number* $C \in \mathbb{Z}$, which can be directly calculated from the occupied eigenstates of the Bloch Hamiltonian $\mathcal{H}(k_x, k_y)$ (see also the lecture notes by Joel Moore[13]).

[12] Again, technically $s \in \mathbb{Z}^+$, but there is a bijection between this set and $\mathbb{Z}$; the usual convention is to just use $\mathbb{Z}$ here.

[13] Joel E Moore. An introduction to topological phases of electrons. *Topol. Aspects Condens. Matter Phys.: Lecture Notes Les Houches Summer School*, 103:1, 2017

## 2.4   3D and higher

By now we know how to play this game. We write down the three-dimensional (3D) Dirac equation

$$h(q_x, q_y, q_z) = \gamma_1 q_x + \gamma_2 q_y + \gamma_3 q_z + \gamma_4, \quad \{\gamma_l, \gamma_m\} = 2\delta_{mn}, \quad (2.38)$$

and explore the space of possible matrices $\gamma_4$ given the choice[14]

$$\gamma_1 = \sigma_x \otimes \mathbb{1}, \quad \gamma_2 = \sigma_y \otimes \mathbb{1}, \quad \gamma_3 = \sigma_z \otimes \mathbb{1}. \quad (2.39)$$

[14] Note that here I have swapped $\sigma_x \leftrightarrow \sigma_z$ compared to the 1D and 2D cases purely out of a certain aesthetic sensibility.

We now have a problem: there is *no* matrix $\gamma_4$ that anticommutes with *all* three matrices $\gamma_1$, $\gamma_2$, $\gamma_3$. (For the experts: we have just rediscovered the fact that 3D Weyl semimetals can only be gapped out by the pairwise annihilation of Weyl points.)

Well, that's bad. We thought we knew how to play this game, but there seems to be something new at every corner. Let's try again: we choose

$$\gamma_1 = \sigma_x \otimes \sigma_z \otimes \mathbb{1}, \quad \gamma_2 = \sigma_y \otimes \sigma_z \otimes \mathbb{1}, \quad \gamma_3 = \sigma_z \otimes \sigma_z \otimes \mathbb{1}, \quad (2.40)$$

where $\mathbb{1}$ now denotes a $M_0/4 \times M_0/4$ identity matrix.

This is much better. We know how to gap this Dirac equation out: inspired by Eq. (2.30) we write

$$\gamma_4 = e^{i(\sigma_0 \otimes \sigma_0 \otimes A + \sigma_0 \otimes \sigma_z \otimes B)} (\sigma_0 \otimes \sigma_x \otimes \mathbb{1}) e^{-i(\sigma_0 \otimes \sigma_0 \otimes A + \sigma_0 \otimes \sigma_z \otimes B)}, \quad (2.41)$$

where I'm using $\sigma_0$ to denote the $2 \times 2$ identity matrix to avoid confusion with $\mathbb{1}$, and $A$ and $B$ are some $M_0/4 \times M_0/4$ Hermitian matrices. We see that the problem is equivalent to the 1D case [Eq. (2.30)], except that there is now an extra $\sigma_0$ in front of every matrix appearing in $\gamma_4$. Correspondingly, the 3D classification is trivial like in 1D, and there are no topological invariants differentiating the possible gapped ground states.

It is now not too hard to see that the classification pattern we have observed just keeps on repeating. We have just verified this explicitly in the 3D case, whose classification (trivial) was the same as in 1D. We have also seen it in the 2D case, where the $\mathbb{Z}$ classification (corresponding to the Chern number) was the same as in 0D (corresponding to the total particle number).

In general, the classification of free fermion SPT phases protected by U(1) symmetry is given by $\mathbb{Z}$ in even dimensions, while it is trivial in odd dimensions. If you like fancy maths, this result is related to a concept called *Bott periodicity*.[15]

[15] If you are interested in a more complete treatment of this result, see here:

Xiao-Gang Wen. Symmetry-protected topological phases in non-interacting fermion systems. *Phys. Rev. B*, 85:085103, Feb 2012. DOI: 10.1103/PhysRevB.85.085103. URL https://link.aps.org/doi/10.1103/PhysRevB.85.085103

## 3 Free fermion topological phases without symmetry

We have seen that the only nontrivial *free* fermion SPTs protected by U(1) charge conservation symmetry exist in zero or two spatial dimensions[1]. If we want to start thinking about interacting fermion systems, this situation is a bit impractical:

(1) The 0D case is obvious – the total number of particles clearly remains a quantized invariant of the ground state even for an interacting many-body Hamiltonian.

(2) The 2D case is a bit too complicated to treat in a pedestrian manner. To understand the effect of interactions, we would need to use fancy quantum field theory methods, which would take a while to introduce.

Let us therefore relax the assumption of U(1) symmetry. This will give us nontrivial fermion topological phases already in 1D, which are much easier to analyse. We here first consider the case without any symmetry, resulting in a $\mathbb{Z}_2$ classification. Then, in Section 4, we use spinless time-reversal symmetry to upgrade from $\mathbb{Z}_2$ to $\mathbb{Z}$.

### 3.1 Majorana fermions and fermion parity

We first discuss the free fermion classification and then study how it is modified by interactions. Dropping the constraint of U(1) symmetry for free fermions, we revert back to Eq. (2.1), repeated here for convenience:

$$H = \sum_{ij} \mathcal{H}_{ij} c_i^\dagger c_j + \Delta_{ij} c_i^\dagger c_j^\dagger + \Delta_{ij}^* c_j c_i. \tag{3.1}$$

This Hamiltonian looks much nicer when expressed in terms of *Majorana fermions*: we introduce the $2N$ operators $\gamma_\alpha$, $\alpha = 1\ldots2N$, so that[2]

$$\gamma_{2i-1} = c_i + c_i^\dagger, \quad \gamma_{2i} = \frac{c_i - c_i^\dagger}{i}. \tag{3.2}$$

The Majorana operators are Hermitian and satisfy the anti-commutation relations

$$\{\gamma_\alpha, \gamma_\beta\} = 2\delta_{\alpha\beta}. \tag{3.3}$$

Using the inverse transformation

$$c_i = \frac{1}{2}(\gamma_{2i-1} + i\gamma_{2i}), \quad c_i^\dagger = \frac{1}{2}(\gamma_{2i-1} - i\gamma_{2i}), \tag{3.4}$$

we can rewrite the Hamiltonian $H$ in the form

$$H = i \sum_{\alpha\beta} A_{\alpha\beta} \gamma_\alpha \gamma_\beta + const., \tag{3.5}$$

[1] In these notes, we restrict to the physically accessible spatial dimensions of $0, 1, 2, 3$. It is amusing to note however that the 4D $\mathbb{Z}$ classification of U(1)-symmetric free fermion SPTs has been realised experimentally using so-called *synthetic dimensions.* See:

Tomoki Ozawa and Hannah M. Price. Topological quantum matter in synthetic dimensions. *Nature Reviews Physics*, 1(5):349–357, 2019

[2] Sorry for overbooking the notation again – these Majorana operators are *not* to be confused with the Dirac $\gamma$-matrices that we had used in Eq. (2.26) and similar.

where one can always choose the constant in such a way that the matrix $A$ is real and anti-symmetric: $A_{\alpha\beta} = -A_{\beta\alpha} \in \mathbb{R}$.[3] From now on, we will drop this constant, because it merely represents some overall energy shift.

Even though this Hamiltonian does not have U(1) charge conservation symmetry, it still has *fermion parity* symmetry: define the operator

$$P = \prod_j (1 - 2c_j^\dagger c_j) = i^N \gamma_{2N}\gamma_{2N-1} \cdots \gamma_2\gamma_1, \qquad (3.6)$$

it is then easy to verify that $[P, H] = 0$ and $P$ is unitary: $P^\dagger P = 1$. Fermion parity symmetry furnishes a $\mathbb{Z}_2$ group because there are only two elements in the group: $P$ and $P^2 = 1$, implying that $P = P^\dagger$ is also Hermitian.

In fact, as mentioned before in Sec. 2, every *local* fermion Hamiltonian – free or interacting – must consist of terms that have an even number of fermion creation and annihilation operators each, meaning it automatically has fermion parity symmetry. In this sense, it is physically impossible to break fermion parity symmetry. Therefore, when a fermion SPT phase as defined in Sec. 1.1 *only* has fermion parity, it should really just be called a topological phase, rather than a SPT phase. However, it is often still helpful to formally treat fermion parity as just another symmetry, essentially a subgroup of the full U(1) charge conservation symmetry[4].

## 3.2 Topological classification in 0D and 1D

Let's quickly speedrun the classification of free fermion SPTs with $\mathbb{Z}_2$ fermion parity symmetry, building on the formalism from Sec. 2. In presence of translational symmetry, it makes sense to define the Fourier transform $A(k)$ of the matrix $A$ from Eq. (3.5) in analogy to the definition of the Bloch Hamiltonian $\mathcal{H}(k)$ in Eq. (2.22): introducing the composite index $\alpha = (R, a)$, where $R$ denotes the unit cell position and $a$ ranges over the Majorana fermions within each unit cell[5], we define

$$A(k)_{ab} = \sum_R e^{-ikR} A_{(R,a),(0,b)}. \qquad (3.7)$$

The full Hamiltonian in Eq. (3.5) then becomes[6]

$$H = i\sum_k A(k)_{ab} \gamma_{k,a} \gamma_{-k,b}, \qquad (3.8)$$

for some new set of momentum space Majorana operators $\gamma_{k,a}$ [we do not need their explicit form right now, but see Eq. (3.16) below for an example of how they are defined in practise].

The anti-symmetry of the real matrix $A$ implies that $A(k)$ satisfies

$$A(k)^\dagger = -A(k), \quad A(k)^* = A(-k). \qquad (3.9)$$

[3] The real anti-symmetric $2N \times 2N$ matrix $A$ now takes over the role of the complex Hermitian $N \times N$ matrix $\mathcal{H}$ that we had used in Eq. (2.4) to study SPTs with U(1) symmetry.

[4] Fermion parity is obtained from U(1) symmetry by restricting to the two choices $\theta = 0, \pi$ in Eq. (2.3).

[5] Note that even if we have a full translational symmetry $i \to i + 1$, there are now *two* Majoranas within each unit cell (corresponding to *a single* fermionic mode). This is because the definition of the Majorana fermions $\gamma_\alpha$ in Eq (3.2) has a fundamental asymmetry between even and odd choices of $\alpha$.
[6] Here we have dropped the constant term of Eq. (3.5).

**0D**   Ground states of different fermion parity eigenvalue $P = \pm 1$ cannot be smoothly connected to one another, so the oD classification is $\mathbb{Z}_2$.

**1D**   As in Sec. 2.2, we need to study a band crossing between some arbitrary number $M_0$ of bands [= eigenstates of the Hermitian matrix $\mathrm{i}A(k)$] at a point in momentum space. Without loss of generality, we can choose this point as $k_0 = 0$.[7] In analogy to Eq. (2.26), the different ways to open a gap will now be described by the continuum model

$$a(k) = \zeta_1 k + \zeta_2, \tag{3.10}$$

where the $M_0 \times M_0$ matrices $\zeta_1$, $\zeta_2$ must be chosen so that $a(k)$ satisfies the same constraints as in Eq. (3.9). Without loss of generality, we can choose

$$\zeta_1 = \mathrm{i}\sigma_z \otimes \mathbb{1}, \tag{3.11}$$

where as before $\mathbb{1}$ denotes the $M_0/2 \times M_0/2$ identity matrix. Note that now $\zeta_1^2 = -1$ squares to minus one, instead of plus one, due to the presence of the imaginary unit i in Eq. (3.5).

What is the most general form of $\zeta_2$ that anti-commutes with $\zeta_1$, also squares to minus one[8], and satisfies the constraints in Eq. (3.9)? To ensure anti-commutation with $\zeta_1$, this matrix should not involve $\sigma_0$ or $\sigma_z$ and therefore takes the general form

$$\zeta_2 = \begin{pmatrix} 0 & \zeta \\ -\zeta^{\mathrm{T}} & 0 \end{pmatrix}, \tag{3.12}$$

where $\zeta \in \mathrm{O}(M_0/2)$ is a $M_0/2 \times M_0/2$ *orthogonal* matrix. Since the orthogonal group has *two* disconnected components (one with $\det \zeta = +1$ and another with $\det \zeta = -1$), we can immediately deduce that the classification is $\mathbb{Z}_2$.

**2D and 3D**   Homework[9].

## 3.3   Kitaev chain and Majorana zeromodes

Okay, enough of the classification math already. We now know that 1D classification is $\mathbb{Z}_2$, and to understand this physically it would be really helpful to have a concrete model at hand. Consider first the Hamiltonian [using the Majorana operators of Eq. (3.2)]

$$H_0 = \mathrm{i} \sum_{j=1}^{N} \gamma_{2j-1} \gamma_{2j} = \sum_{j=1}^{N} \left( 2c_j^\dagger c_j - 1 \right) = \sum_{j=1}^{N} \left( 2\hat{n}_j - 1 \right). \tag{3.13}$$

This Hamiltonian decomposes as a sum of commuting terms over different sites $j = 1 \ldots N$. Since the eigenvalues of $\hat{n}_j$ are 0 and 1,

[7] This choice was not important in the case of U(1) symmetry, but now we have to deal with the condition in Eq. (3.9). Whenever we have a gap closing at a finite $k_0 \neq 0$, this condition will enforce a gap closing also at $-k_0$. It therefore makes sense to first move the two gap closings to the same momentum $k_0 = 0$ (or $k_0 = \pi$). We then end up with twice the number of bands, but only need to consider a single crossing point.

[8] see the text below Eq. (2.26)

[9] [spoiler] It's $\mathbb{Z}$ in 2D and trivial in 3D. Unlike the case of U(1) symmetry, the 2D (3D) classification is now different from oD (1D). In fact, breaking U(1) symmetry upgrades the Bott periodicity that we had mentioned at the end of Sec. 2.4 to mod 8 instead of mod 2.

the ground state = minimal energy state of $H_0$ is given by the choice $\hat{n}_j = 0$ on all sites: the ground state is simply the fermionic vacuum

$$|\text{GS}_0\rangle = |0\rangle. \tag{3.14}$$

This state is quite boring, but let us file it away for later use as a trivial reference state (Sec. 1.1)[10].

Can we think about a Hamiltonian that is a bit more interesting that this? Ideally we'd like to construct the $\mathbb{Z}_2$-nontrivial free fermion SPT phase that we had discovered in Sec. 3.2. Kitaev[11] had the insight to study the Hamiltonian

$$H_1 = i \sum_{j=1}^{N} \gamma_{2j}\gamma_{2j+1}, \tag{3.15}$$

which differs from $H_0$ by a translation by just one Majorana mode. This Hamiltonian – a special version of the celebrated "Kitaev chain" Hamiltonian – has a much more messy form when expressed in terms of the $c_j^\dagger$, $c_j$ operators, which we will not need here. In particular, $H_1$ will now contain pairing terms $\Delta$ when expressed in the form of Eq. (3.1), and therefore clearly breaks U(1) symmetry.

We will now show that the ground state of $H_1$ indeed realises the nontrivial SPT phase. Using the Fourier transforms

$$\gamma_{2j-1} = \frac{1}{\sqrt{L}} \sum_k e^{-ikj}\gamma_{k,1}, \quad \gamma_{2j} = \frac{1}{\sqrt{L}} \sum_k e^{-ikj}\gamma_{k,2}, \tag{3.16}$$

we can transform $H_1$ to momentum space[12]:

$$H_1 = i \sum_k e^{ik}\gamma_{k,2}\gamma_{-k,1} = \frac{i}{2} \sum_k \left( e^{ik}\gamma_{k,2}\gamma_{-k,1} - e^{-ik}\gamma_{k,1}\gamma_{-k,2} \right), \tag{3.17}$$

so that from Eq. (3.8) we can read off

$$A_1(k) = \frac{1}{2} \begin{pmatrix} 0 & -e^{-ik} \\ e^{ik} & 0 \end{pmatrix}. \tag{3.18}$$

As a consistency check, we note that this matrix satisfies the constraints of Eq. (3.9). Conversely, another short calculation shows that the momentum space matrix $A_0(k)$ associated with $H_0$ in Eq. (3.13) is $k$-independent and reads

$$A_0(k) = \frac{1}{2} \begin{pmatrix} 0 & 1 \\ -1 & 0 \end{pmatrix}. \tag{3.19}$$

We now define a linear interpolation between the two Hamiltonians $H_0$ and $H_1$ for a parameter $t \in [0,1]$:

$$H_t = (1-t)H_0 + tH_1, \tag{3.20}$$

[10] Note that for U(1)-preserving SPTs, this would be a very bad choice of a trivial reference state: the Hilbert space at fixed particle number $\hat{N} = 0$ is one-dimensional and *only* contains the state $|0\rangle$. However, the Hilbert space of even fermion parity $P = 1$, which contains $|0\rangle$, is exponentially large (its dimension is $2^{N-1}$).

[11] Alexei Kitaev. Unpaired majorana fermions in quantum wires. *Physics-Uspekhi*, 44(10S):131, oct 2001. DOI: 10.1070/1063-7869/44/10S/S29. URL https://dx.doi.org/10.1070/1063-7869/44/10S/S29

[12] As already remarked earlier, we need *two* different momentum space Majorana operators $\gamma_{k,1}$ and $\gamma_{k,2}$ here because the definition of the $\gamma_\alpha$'s in Eq. (3.2) is not translationally symmetric under the substitution $\alpha \to \alpha + 1$.

so that $H_t$ corresponds to the momentum space matrix

$$
A_t(k) = (1-t)A_0(k) + tA_1(k) = \frac{1}{2}\begin{bmatrix} 0 & (1-t) - te^{-ik} \\ -(1-t) + te^{ik} & 0 \end{bmatrix}
$$

$$
= \frac{1}{2}\left[(1-t - t\cos k)\, i\sigma_y + t\sin k\, i\sigma_x\right].
$$

(3.21)

The eigenvalues of the *Hermitian* matrix $iA_t(k)$ are given by

$$
\epsilon_\pm(t,k) = \pm\frac{1}{2}\sqrt{(1-t-t\cos k)^2 + (t\sin k)^2}.
\tag{3.22}
$$

In particular, we see that at $t = 1/2$, there is a gap closing at $k = 0$. Expanding $A_t(k)$ to linear order in $m = (t - 1/2)$ and $k = (k - 0)$ about this gap closing, we obtain the Dirac matrix

$$
a(k,m) = \frac{1}{2}\left[-2m\, i\sigma_y + \left(m + \frac{1}{2}\right)k\, i\sigma_x\right]
$$

$$
= \frac{1}{4}(1 + 2m)k\, i\sigma_x - m\, i\sigma_y.
$$

(3.23)

Up to permutation of the Pauli matrices $\sigma_x \leftrightarrow \sigma_z$ and a rescaling of the different terms, $a(k,m)$ is of the form of the continuum model that we had considered below Eq (3.10) when restricting to $M_0 = 2$ bands. In particular, the orthogonal matrix $\zeta$ that appears in Eq. (3.12) is then simply given by $\zeta = -\text{sign}(m)$, as long as we rescale it to have unit norm so as to give a bona fide orthogonal "matrix". But this means that as we tune across the gap closing point $m = 0$, at which the sign of $m$ changes, the sign of $\det\zeta = -\text{sign}(m)$ also changes sign. Correspondingly, this is a phase transition between the two inequivalent 1D topological phases that we had identified in Sec. 3.2. The phase at $m < 0$, exemplified by the ground state $H_0$, is our trivial reference state $|0\rangle$ [Eq. (3.14)]. But then the phase with $m > 0$, exemplified by the ground state of $H_1$, *must* realise the nontrivial SPT phase.

*Some generalities on the SPT state in PBC*   We now study the ground states of $H_0$ and $H_1$ in periodic boundary conditions (PBC) and later also open boundary conditions (OBC). In Eqs. (3.13) and (3.15), PBC are defined by setting $\gamma_{2N+1} \equiv \gamma_1$ (and $\gamma_{2N+2} \equiv \gamma_2$), so that $c_{N+1}^\dagger \equiv c_N$. OBC are defined by removing all terms coupling between site $i = N$ and $i = N+1$ from the Hamiltonian. Note that $H_0$ is the same in OBC and PBC, and so its ground state $|GS_0\rangle = |0\rangle$ is the same. On the other hand, $H_1$ in PBC contains an extra term $+i\gamma_{2N}\gamma_1$ that is absent in OBC. This means that the ground state(s) of $H_1$ will differ between PBC and OBC.

Consider the *unitary* "Majorana translation" operator $\tau$ that is defined by

$$
\tau\gamma_\alpha\tau^\dagger = \gamma_{\alpha+1} \quad (\gamma_{2N+1} \equiv \gamma_1).
\tag{3.24}
$$

A brief calculation shows that this operator always[13] anti-commutes with the fermion parity operator $P$ that we had defined in Eq. (3.6):

$$\tau P \tau^\dagger = -P. \tag{3.25}$$

[13] irrespective of the system size $L$

In PBC, $H_1$ and $H_0$ are related to each other by $\tau$:

$$H_1 = \tau H_0 \tau^\dagger. \tag{3.26}$$

This means that their spectrum is the same and the ground state $|GS_1\rangle$ of $H_1$ can be obtained from that of $H_0$ in Eq. (3.14):

$$|GS_1\rangle = \tau |GS_0\rangle = \tau |0\rangle. \tag{3.27}$$

Importantly, Eq. (3.25) implies that $|GS_1\rangle$ has opposite fermion parity to $|GS_0\rangle$, and therefore must have odd parity:

$$P |GS_1\rangle = - |GS_1\rangle. \tag{3.28}$$

The state $|GS_1\rangle$ looks pretty innocuous, but as we have shown above it is a nontrivial SPT state, in the sense that it cannot be continuously connected to the ground state of $H_0$.[14]

***Explicit form of the SPT state in PBC***   Let's now construct the explicit ground state of $H_1$ in PBC. We will avoid using the operator $\tau$ for this, as we want a construction that can be straightforwardly generalised also to OBC later on, where Eq. (3.26) does not hold. Consider the operator

$$P_j = \frac{1}{2} \left(1 - i\gamma_{2j}\gamma_{2j+1}\right). \tag{3.29}$$

This is a projection operator because it satisfies $P_j^2 = P_j$ and it has eigenvalues $0, 1$. In fact, $P_j$ projects onto the eigenvalue $(-1)$ eigenstate of the operator $i\gamma_{2j}\gamma_{2j+1}$ that appears in $H_1$:

$$(i\gamma_{2j}\gamma_{2j+1})P_j = \frac{1}{2}(i\gamma_{2j}\gamma_{2j+1} - 1) = -P_j. \tag{3.30}$$

Since all terms in $H_1$ commute with each other, we could naively try to build the PBC ground state of $H_1$ as the mutual $(-1)$-eigenvalue eigenstate of all these terms, in the following way:

$$|GS_1\rangle = \frac{1}{\mathcal{N}} \prod_{j=1}^{N} P_j |0\rangle, \tag{3.31}$$

where $\mathcal{N}$ is a normalisation factor. However, this ansatz is subtly wrong: We have seen in Eq. (3.28) that $|GS_1\rangle$ has odd fermion parity, but since all $P_j$ have even parity this requirement is not satisfied by the above equation[15]. Instead, we must use an ansatz like

[14] The simplest way to see that $|GS_1\rangle$ and $|GS_0\rangle$ cannot be continuously connected is by noting that they have a different fermion parity eigenvalue. At this point, however, you could make a subtle objection based on the discussion right at the beginning of Sec. 2.2: since the fermion parity eigenvalue differentiates between ground states already in 0D, we really shouldn't use it to argue why the 1D topological phase cannot be smoothly deformed to *any* 1D trivial reference state. (Using parity eigenvalues would just reproduce the 0D classification, and not rely on 1D locality in any way.) In fact, the 1D topological phase *can* be made trivial, *without changing the fermion parity of the ground state*, as soon as nonlocal couplings are allowed. For example, consider the alternative trivial reference state $|triv\rangle = c_1^\dagger |0\rangle$ that has odd fermion parity. The ground state of the Hamiltonian $H_t = -t |triv\rangle \langle triv| - (1-t) |GS_1\rangle \langle GS_1| + \Delta t(1-t)(|GS_1\rangle \langle triv| + |triv\rangle \langle GS_1|)$, where $\Delta \ll 1$ is some small real coupling term, interpolates between $|GS_1\rangle$ at $t=0$ and $|triv\rangle$ at $t=1$ while keeping the gap open. However, this Hamiltonian is terribly nonlocal!

[15] This observation leads us to an apparent paradox: by construction, Eq. (3.31) should be an eigenstate of $H_1$ with energy $-N$. This is exactly the ground state energy of $H_0$, and therefore also that of $H_1$ in PBC as they are related by a unitary [Eq. (3.26)]. The only way in which the paradox could be resolved is if $\prod_{j=1}^{N} P_j |0\rangle = 0$ was in fact zero and therefore not a normalisable ground state. You can check by explicit computation for small $N$ that this is indeed the case.

$$|\text{GS}_1\rangle = \frac{1}{\mathcal{N}} \left( \prod_{j=1}^{N} P_j \right) c_1^\dagger |0\rangle , \tag{3.32}$$

where the operator $c_1^\dagger$ is an arbitrary choice that makes the fermion parity come out as odd. Using this form of the ground state and Eq. (3.30), we find

$$H_1 |\text{GS}_1\rangle = -N |\text{GS}_1\rangle , \tag{3.33}$$

which gives the ground state energy $-N$ in PBC as expected. Importantly, the ground state is unique – the lowest lying excitation is to flip the eigenvalue of a single term $i\gamma_{2i}\gamma_{2i+1}$ in $H_1$ by replacing $P_i$ with $(1 - P_i)$ for some choice of $i$ in Eq. (3.31), which incurs an energy penalty of $1 - (-1) = 2$. Moreover, the ground state preserves fermion parity symmetry, in the sense that it is an eigenstate of the fermion parity operator $P$ of eigenvalue $-1$. Therefore, $|\text{GS}_1\rangle$ satisfies all the defining features of a SPT state, see Sec. 1.1.

Using Eq. (3.32), we can now readily compute the explicit form of the PBC ground state for a given $N$. For example, for a single site $N = 1$, we obtain

$$|\text{GS}_1\rangle = c_1^\dagger |0\rangle , \tag{3.34}$$

while for $N = 2$ we obtain[16]

$$|\text{GS}_1\rangle = \frac{1}{\sqrt{2}} \left( c_1^\dagger + c_2^\dagger \right) |0\rangle . \tag{3.35}$$

Note that the apparent asymmetry between site 1 and 2 that was present in our ansatz, Eq (3.32), has vanished as it should: since $|\text{GS}_1\rangle$ is by construction the unique ground state of a translationally symmetric Hamiltonian, it must preserve translational symmetry and not favour one site over another. This reassures us that the arbitrary choice we had made by using $c_1^\dagger$ instead of some other $c_j^\dagger$ in Eq (3.32) is without loss of generality.

*Degenerate ground states in OBC*   What happens to the ground state of $H_1$ in OBC? To recall, in OBC we cut the bound between site $N$ and 1, creating a left and right end point of the chain. Removing this bond from Eq. (3.15), $H_1$ becomes

$$H_1 = i \sum_{j=1}^{N-1} \gamma_{2j}\gamma_{2j+1} \quad \text{(OBC)}, \tag{3.36}$$

which is still a sum of commuting terms and so we can immediately write down *one* ground state, which is the same as before[17]:

$$|\text{GS}_1\rangle = \frac{1}{\mathcal{N}} \left( \prod_{j=1}^{N} P_j \right) c_1^\dagger |0\rangle . \tag{3.37}$$

[16] We cannot naively generalise to general $N$ from these two examples. The ground state is not always just a superposition of the form $\sum_i c_i^\dagger |0\rangle$. Starting with $N = 3$, it will also include products such as $c_1^\dagger c_2^\dagger c_3^\dagger |0\rangle$, and so on.

[17] Note that even though we are in OBC now, we keep the definition $P_N = (1 - i\gamma_{2N}\gamma_1)/2$. OBC is a statement about what terms appear in the Hamiltonian, which does not translate simply to the terms that may appear in the ground state. In fact, we quite like the identification $c_{N+1} \equiv c_1$, and will keep it around for convenience.

This state now has an energy

$$H_1 \left| \text{GS}_1 \right\rangle = -(N-1) \left| \text{GS}_1 \right\rangle. \tag{3.38}$$

In contrast to the PBC case, however, we note that in OBC we can also write down a second ground state

$$\left| \text{GS}_1' \right\rangle = c_{\text{NL}}^{\dagger} \left| \text{GS}_1 \right\rangle, \tag{3.39}$$

where we have introduced the *nonlocal* fermion mode $c_{\text{NL}}^{\dagger}$ defined by

$$c_{\text{NL}}^{\dagger} = \frac{1}{2}(\gamma_{2N} - i\gamma_1). \tag{3.40}$$

$\left| \text{GS}_1' \right\rangle$ and $\left| \text{GS}_1 \right\rangle$ have the *same* energy due to the absence of the operators $\gamma_{2N}$ and $\gamma_1$ in Eq. (3.36), implying $[c_{\text{NL}}^{\dagger}, H_1] = 0$ in OBC. The two states are physically inequivalent, as we can check by computing their overlap

$$\left\langle \text{GS}_1' | \text{GS}_1 \right\rangle = 0, \tag{3.41}$$

where we have used $c_{\text{NL}} \left| \text{GS}_1 \right\rangle = 0$, which can also be used to show that $\left\langle \text{GS}_1' | \text{GS}_1' \right\rangle = 1$ is properly normalised. We have therefore found that $H_1$ has *two* degenerate ground states in OBC, and it is not hard to convince yourself that all its remaining eigenstates are gapped from these two and have higher energies.

Some words to drape around this result: We say that the OBC Hamiltonian $H_1$ has two *Majorana zeromodes* $\gamma_1$ and $\gamma_{2N}$, which means that they both commute with $H_1$ and therefore do not change the energy. We already know from Eq. (3.4) that two Majoranas make one physical fermion – it's just that now the Majoranas are separated by the full chain and so we can only build a *nonlocal* fermion $c_{\text{NL}}^{\dagger}$ whose occupation number operator commutes with the Hamiltonian:

$$[\hat{n}_{\text{NL}}, H_1] = 0, \quad \hat{n}_{\text{NL}} = c_{\text{NL}}^{\dagger} c_{\text{NL}} = 1 - P_N. \tag{3.42}$$

We now see how the two OBC ground states differ in physical terms: one has the nonlocal fermion mode empty:

$$\hat{n}_{\text{NL}} \left| \text{GS}_1 \right\rangle = 0, \tag{3.43}$$

while the other degenerate ground state has the nonlocal fermion mode occupied:

$$\hat{n}_{\text{NL}} \left| \text{GS}_1' \right\rangle = \left| \text{GS}_1' \right\rangle. \tag{3.44}$$

## 3.4   Bulk-boundary correspondence

The twofold OBC ground state degeneracy of $H_1$ is a unique feature of the topological phase. Consider in contrast the ground state of $H_0$, which as we have already noted above takes the same form in PBC

and OBC (it is simply the fermionic vacuum): $|GS_0\rangle = |0\rangle$. This state is trivially unique and gapped, because the PBC and OBC versions of $H_0$ are one and the same[18]. Therefore, we can use the OBC ground state degeneracy to uniquely diagnose the SPT phase. We speak of a *bulk-boundary correspondence*: the topologically nontrivial bulk of the SPT phase supports protected zero-energy edge excitations[19].

Importantly, the bulk-boundary correspondence is a property of the entire SPT *phase*, not just of a single Hamiltonian such as $H_1$: Since the two degenerate ground states of $H_1$ are related to each other by the nonlocal operator $c_{NL}^\dagger$, their energies cannot be split by *any* local perturbation. Consider for example an OBC version of the interpolation of Hamiltonians $H_t$ in Eq. (3.20) that tunes between the bulk Hamiltonians $H_0$ and $H_1$: as long as we are in the SPT phase ($t > 1/2$), the OBC ground state degeneracy remains perfect in the thermodynamic limit[20]. On the other hand, in the trivial phase, the ground state is unique in both PBC and OBC.

The sentence highlighted in blue above is not very rigorous and only gives us an illusion of understanding. We really need a

***Slightly more rigorous proof of the bulk-boundary correspondence***
Comparing Eq. (3.36) with Eq. (3.5), we can read off

$$A = \frac{1}{2} \sum_{j=1}^{N-1} \left( |2j\rangle \langle 2j+1| - |2j+1\rangle \langle 2j| \right), \qquad (3.45)$$

where we are using the first-quantised notation $|\alpha\rangle$ to denote the $\alpha$-th unit vector of a $2N$-dimensional auxiliary Hilbert space[21]. The Hermitian matrix $iA$ has two zero-energy[22] eigenvectors $iA|\psi_m\rangle = 0$, $m = 1, 2$, where

$$|\psi_1\rangle = |1\rangle, \quad |\psi_2\rangle = |2N\rangle. \qquad (3.46)$$

These two eigenvectors correspond to our Majorana zeromodes, localised at the ends of the system. All other eigenvalues of the $iA$ appear in pairs $\pm 1/2$ and are therefore gapped from zero energy.

Recall that we had used the many-body Majorana zeromode operators $\gamma_1$ and $\gamma_{2N}$ to construct a nonlocal fermion operator that commutes with the Hamiltonian and therefore gave rise to a two-fold ground state degeneracy in OBC. We can trace back the vanishing commutators $[\gamma_1, H] = [\gamma_{2N}, H] = 0$ to the fact that the states $|\psi_m\rangle$ in Eq. (3.46) are zero-energy eigenvectors of $iA$.[23] To prove the bulk boundary correspondence in general, we therefore need to show that the zeromodes of $iA$ are stable as long as we do not close the gap to the rest of the spectrum.

The general eigenvalue equation for $iA$ reads

$$iA|\psi\rangle = E|\psi\rangle, \qquad (3.47)$$

[18] This comes from the fact that $H_0$ does not involve any coupling between the different lattice sites $i = 1 \dots N$, in particular not between site $N$ and 1.

[19] In our concrete example, these excitations are the Majorana zeromode operators $\gamma_1$ and $\gamma_{2N}$.

[20] For a finite system size $N$, the degeneracy is split by an amount that is exponentially small in $N$.

[21] This is just notation to make our life easier. The matrix elements that appear in Eq. (3.5) are given by $A_{\alpha\beta} = \langle\alpha|A|\beta\rangle$, and we could perform the same analysis using explicit index notation.

[22] Our use of the word "energy" is a bit loose here: This is the energy of the single-particle excitations described by the matrix $iA$, *not* the energy of any many-body (ground) states. The distinction between two kinds of energy, single-particle and many-body, is important but often not mentioned explicitly. For example, in the term "Majorana zeromode", the "zero" really indicates that these modes have zero *single-particle* energy, meaning they are zero-energy eigenvectors of the matrix $iA$. In the many-body picture, Majorana zeromodes manifest themselves not in a many-body state at zero energy, but in the many-body ground state *degeneracy* that we have been talking about earlier.

[23] Homework: prove this :)

where $E$ is some real energy. Since $A$ is a real matrix, a complex conjugation of both sides results in

$$i A \ket{\psi^*} = -E \ket{\psi^*}, \qquad (3.48)$$

where by $\ket{\psi^*}$ we denote the vector whose coefficients in the basis $\ket{\alpha}$, $\alpha = 1 \ldots 2N$, are the complex conjugates of those of $\ket{\psi}$. This observation implies that the eigenvalues of $iA$ always come in pairs $E, -E$, unless of course when $E = 0$ where $\ket{\psi} = \ket{\psi^*}$ may be the same vector.

Importantly, complex conjugation does not change the spatial profile of an eigenvector. Therefore, no smooth deformation of $iA$ that preserves the bulk gap can move the states $\ket{\psi_m}$ of Eq. (3.46) away from zero energy: assume that this was possible, meaning we are able to shift for example $\ket{\psi_1}$ to a small finite energy $0 \neq E_1 \ll 1/2$. Then Eq. (3.48) implies that there should be another state at energy $-E_1$ *with the same spatial profile*, meaning it would also need to be localised at the left edge of the chain. This is a contradiction, because we only started out with a single zero eigenvalue originating from the left edge, and all eigenvalues and eigenstates must vary smoothly[24] with the matrix $iA$. Therefore, to get rid of the zero-modes, we must either couple $\ket{\psi_1}$ with $\ket{\psi_2}$ – this violates locality – or we must close the bulk gap, at which point the eigenvectors $\ket{\psi_1}$ and $\ket{\psi_2}$ become delocalised so that they can be coupled without breaking locality.

In conclusion, the zero-energy end states of the matrix $iA$ – and by extension the OBC ground state degeneracy of the Kitaev chain Hamiltonian in Eq. (3.15) – must persist as a unique signature of the SPT phase away from the fine-tuned limit of Eq. (3.36), and throughout the entire phase.

It is now easy to see that *two* Kitaev chains are trivial: we will then have *two* zero-energy eigenvectors of $iA$ at the *same* edge, and adding a local coupling between them to the Hamiltonian will push them out to finite and opposite energies.

***Confirming the $\mathbb{Z}_2$ classification***    We have seen that $H_0$ in Eq. (3.13) gives a trivial reference ground state, while $H_1$ in Eq. (3.15) gives a topological ground state that has a two-fold degeneracy in OBC. From Sec. 3.2 we furthermore know that the classification should be $\mathbb{Z}_2$, meaning that there are only *two* inequivalent phases – one trivial, the other SPT. Correspondingly, if we stack two Kitaev chains, we should be able to fully gap out the OBC degeneracy.

We can analyse this situation graphically. Let's first draw $H_1$ in Fig. 3.4 below. We see the two unpaired Majorana operators $\gamma_1$ and $\gamma_{2N}$ that are responsible for the ground state degeneracy.

[24] up to gauge transformations, of course, which also do not change the spatial profile of a state

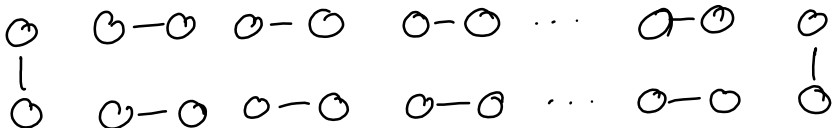

Figure 1: The OBC Kitaev chain Hamiltonian $H_1$ from Eq. (3.36). Each circle represents a Majorana operator $\gamma_\alpha$. Each bar represents a coupling term of the form $i\gamma_\alpha\gamma_\beta$ that appears in the Hamiltonian.

Consider now two copies of $H_1$. They look like Fig. 3.4.

Figure 2: Two copies of $H_1$.

Without doing anything in the bulk – and thereby without closing the bulk gap – we can now include local couplings between the edge Majoranas as shown in Fig. 3.4.

Figure 3: Two copies of $H_1$ with local edge couplings.

But this last situation is clearly equivalent to a single copy of $H_1$, on a lattice with twice as many sites, *with PBC rather than OBC*. Since we already know that $H_1$ in PBC has a unique gapped ground state, this means that we were able to fully lift the degeneracy. This argument is enough to show that two Kitaev chains are topologically trivial, confirming the $1 + 1 = 0 \leftrightarrow \mathbb{Z}_2$ nature of the SPT phase.

## 4 Free fermion SPTs with spinless time-reversal symmetry

The $\mathbb{Z}_2$ classification of the Kitaev chain is a bit limiting and we want more. How can we get more? One simple way is to add spinless time-reversal symmetry.

Time-reversal symmetry is represented by an operator $T$ that commutes with our spinless microscopic fermions $c_i^\dagger$ – after all, time-reversing a static electron that's just sitting at a position $i$ in space should still result in an electron sitting at the same position.[1] We therefore have

$$T c_i^\dagger T^{-1} = c_i^\dagger. \tag{4.1}$$

At the same time, if we have a moving electron – with a momentum $k$ such as the electron $c_k^\dagger$ from Eq. (2.19)[2] – then $T$ should better reverse it's momentum as time now flows backwards:

$$T c_k^\dagger T^{-1} = c_{-k}^\dagger. \tag{4.2}$$

But since $c_k^\dagger = \frac{1}{\sqrt{L}} \sum_j e^{ikj} c_j^\dagger$ this must mean that $T$ is an anti-unitary operator[3] that anti-commutes with imaginary numbers, $T i T^{-1} = -i$.

What does that mean for the Majorana operators? Recall from Eq. (3.4) that

$$c_i = \frac{1}{2}(\gamma_{2i-1} + i\gamma_{2i}), \tag{4.3}$$

so that from $T c_i T^{-1} = c_i$ we must have

$$T \gamma_{2i-1} T^{-1} = \gamma_{2i-1}, \quad T \gamma_{2i} T^{-1} = -\gamma_{2i}. \tag{4.4}$$

This is suprising! $T$ commutes with the odd Majoranas but not with the even Majoranas. But since the Majorana are only "fake" variables that we had introduced to rewrite the Hamiltonian, we should not be too worried about this result.

How do we couple a bunch of Majoranas in a non-interacting Hamiltonian? The only kind of term available is a coupling of the form $i\gamma_\alpha \gamma_\beta$, examples of which already appear in $H_0$ in Eq. (3.13) and $H_1$ in Eq. (3.15). We now see that this term is compatible with time-reversal symmetry only as long as it couples between even and odd Majoranas, and it is disallowed (does not commute with $T$) for even-even or odd-odd pairings. Since this constraint is satisfied for all terms in $H_0$ and $H_1$, we have $[T, H_0] = [T, H_1] = 0$, so that both Hamiltonians already preserve time-reversal symmetry out of the box.

How about 2 copies of $H_1$? Previously, we had shown that these are trivial. But now the rules of the game have changed, as we need to preserve time-reversal symmetry when coupling them. And why

[1] Contrast this with spinful time-reversal that sends $c_\uparrow^\dagger \to c_\downarrow^\dagger$. We do not consider electronic spin in these notes. You might ask how we can get away with this, considering that the spin-statistics theorem of quantum field theory states that fermions must always have half-integer spin. We get away with it because we're doing condensed matter physics here, where Lorentz invariance – a central assumption of the theorem – is broken.

[2] Assuming there are no intra-unit cell degrees of freedom so that $a = 1$ is the only choice and we can suppress this index.

[3] Wigner has shown that *all* symmetries in quantum mechanics must be either unitary or anti-unitary. For a lucid proof, see:

Steven Weinberg. *The Quantum Theory of Fields.* Cambridge University Press, 1995

even stop at 2? Why not consider $n$ copies of $H_1$ already? Let's see where this gets us. We introduce $n$ copies of our fermionic operators, $c_i^{(\lambda)}$, $i = 1 \ldots N$, $\lambda = 1 \ldots n$, with associated Majorana modes $\gamma_\alpha^{(\lambda)}$, $\alpha = 1 \ldots 2N$, $\lambda = 1 \ldots n$. We then consider the Hamiltonian

$$H = \mathrm{i} \sum_{\lambda=1}^{n} \sum_{j=1}^{N} \gamma_{2j}^{(\lambda)} \gamma_{2j+1}^{(\lambda)}, \tag{4.5}$$

which for now is just a direct sum of a $n$ Kitaev chain Hamiltonians $H_1$. In OBC, this Hamiltonian will have $n$ unpaired Majorana modes at each end, which are given by $\gamma_1^{(\lambda)}$ on the left end and $\gamma_{2N}^{(\lambda)}$ on the right end. What's interesting is that now time-reversal symmetry forbids us from coupling these modes: from Eq. (4.4), we have the constraint

$$T \gamma_{2i-1}^{(\lambda)} T^{-1} = \gamma_{2i-1}^{(\lambda)}, \quad T \gamma_{2i}^{(\lambda)} T^{-1} = -\gamma_{2i}^{(\lambda)}. \tag{4.6}$$

This means that all *local* quadratic coupling terms of the form

$$\mathrm{i} \gamma_1^{(\lambda)} \gamma_1^{(\lambda')}, \quad \mathrm{i} \gamma_{2N}^{(\lambda)} \gamma_{2N}^{(\lambda')}, \tag{4.7}$$

which do not mix between the left and right-hand edge, are now disallowed.[4] Correspondingly, we cannot gap out the ground state degeneracy of any integer number $n$ of Kitaev chains, suggesting that the free fermion classification is now upgraded from $\mathbb{Z}_2$ to $\mathbb{Z}$.

Again, our argument for the stability of this $2^n$-fold ground state degeneracy in presence of time-reversal symmetry is a bit hand-wavy. We'd be much happier if we had a

***Slightly more rigorous proof of the bulk-boundary correspondence in presence of time-reversal symmetry***   We again consider the constraints on the Hermitian matrix $\mathrm{i}A$ [Eq. (3.5)] and its eigenvalues. Generalising the first-quantised notation of Eq. (3.45) to include the index $\lambda$ in the auxiliary basis $|\lambda, \alpha\rangle$, $\lambda = 1 \ldots n$, $\alpha = 1 \ldots 2N$, we can read off from Eq. (4.5) that

$$A = \frac{1}{2} \sum_{\lambda=1}^{n} \sum_{j=1}^{N-1} \left( |\lambda, 2j\rangle \langle \lambda, 2j+1| - |\lambda, 2j+1\rangle \langle \lambda, 2j| \right). \tag{4.8}$$

We now need to show that an *arbitrary* number of edge-localised eigenvectors of $\mathrm{i}A$ with zero eigenvalue remains stable and pinned to zero energy. The time-reversal symmetry constraint of Eq. (4.4) translates to the requirement that

$$U_T (\mathrm{i}A)^* U_T^\dagger = \mathrm{i}A, \tag{4.9}$$

where the complex conjugation derives from the fact that $T$ is anti-unitary and $U_T$ denotes the unitary matrix

$$U_T = \sum_{\lambda=1}^{n} \sum_{i=1}^{N} \left( |\lambda, 2i-1\rangle \langle \lambda, 2i-1| - |\lambda, 2i\rangle \langle \lambda, 2i| \right). \tag{4.10}$$

[4] For example, $T \mathrm{i} \gamma_1^{(1)} \gamma_1^{(2)} T^{-1} = -\mathrm{i} \gamma_1^{(1)} \gamma_1^{(2)}$. While $T$ commutes with both Majorana operators, it anti-commutes with the i that is necessary to make this term Hermitian. This is exactly the term that we had used in Fig. 3.4 to show that the ground state degeneracy is lifted. Since the perturbation is now forbidden by $T$, the ground state degeneracy of two Kitaev chains cannot be trivialised as long as time-reversal is preserved.

Using that $A$ is a real matrix, Eq. (4.9) becomes

$$U_T A U_T^\dagger = -A, \qquad (4.11)$$

meaning that $U_T$ anti-commutes with $A$. Since applying time reversal symmetry twice should amount to the identity operation[5] and $U_T$ has real coefficients in the basis $|\lambda, \alpha\rangle$, we furthermore have $U_T^2 = 1$. Correspondingly, the possible eigenvalues of $U_T$ are given by $\pm 1$.

Consider now $n$ copies of the Kitaev chain with associated zero-modes

$$iA |\psi_\lambda^{(L)}\rangle = 0 \qquad (4.12)$$

localised to the *left* end of the chain. (The discussion that follows can be equally well applied to the right edge.) Since these states have zero eigenvalue under $iA$, we can choose them as simultaneous eigenstates also of the operator $U_T$.[6] And because we have just copied the chain $n$ times, their eigenvalues under $U_T$ must all be the same. Let's fix them to be equal to $+1$ without loss of generality[7], so that

$$U_T |\psi_\lambda^{(L)}\rangle = |\psi_\lambda^{(L)}\rangle. \qquad (4.13)$$

We now do a proof by contradiction. Let us assume that it is possible to fully gap out the $n$ states. We have already seen in Eq. (3.48) that $n$ needs to be even for this to be even in principle okay, because the eigenvalues of $iA$ always come in pairs $E, -E$. This means that we will have

$$iA |\psi_\lambda^{(L)}\rangle = \begin{cases} +\epsilon |\psi_\lambda^{(L)}\rangle, & \lambda = 1, \ldots, \frac{n}{2} \\ -\epsilon |\psi_\lambda^{(L)}\rangle, & \lambda = \frac{n}{2} + 1, \ldots, n, \end{cases} \qquad (4.14)$$

where $0 < \epsilon \ll 1/2$ is some small finite energy. The constraint in Eq. (4.11) now ensures that $U_T$ *flips* the energy of a given eigenstate, so that the $n \times n$ matrix

$$M_{\lambda, \lambda'} = \langle \psi_\lambda^{(L)} | U_T | \psi_{\lambda'}^{(L)} \rangle \qquad (4.15)$$

must be *traceless*, $\mathrm{tr} M = 0$. But since the trace of this matrix cannot change discontinuously as we start to gap out the zeromodes, this is a contradiction with Eq. (4.13): Initially, $M$ has finite trace $\mathrm{tr} M = n$, because all $U_T$ eigenvalues in the zero-energy subspace are initially $+1$. The trace of $M$ cannot suddenly jump to zero as $iA$ is smoothly varied, meaning that all $n$ zero-energy eigenvalues remain pinned at zero energy and cannot move away. This proves that any integer number $n$ of Majorana zeromodes remain protected as long as they have the same $U_T$ eigenvalue. In conclusion, we have confirmed that the classification in presence of spinless time-reversal symmetry is $\mathbb{Z}$, which corresponds to the number of stable Majorana zeromodes at each end of the 1D system.

[5] This is actually a subtle constraint that is peculiar to *spinless* time reversal symmetry. For a single spin-1/2 electron, time reversal squares to $-1$ instead of $+1$. (This is similar to the statement that the wavefunction of a spin-1/2 particle picks up a minus sign when we rotate it by an angle of $2\pi$ in space.) For two spin-1/2 electrons, the two minus signs cancel out, so that time reversal again squares to $+1$. Correspondingly, for a system of many spinful electrons, we have $T^2 = P$ where $P$ is the fermion parity operator defined in Eq. (3.6). We here do not pursue this option, as it does not lead to a $\mathbb{Z}$ classification. Instead, we restrict to spinless time reversal where $T^2 = 1$ always holds. The different choices of $T^2$ are also discussed at the beginning of the following lecture notes:
Andrei Bernevig and Titus Neupert. Topological superconductors and category theory. *Lecture Notes of the Les Houches Summer School: Topological Aspects of Condensed Matter Physics*, pages 63–121, 2017

[6] Notably this is *not* possible for the other eigenstates of $iA$ that have finite energy, because $U_T$ anti-commutes with $A$ rather than commutes.

[7] We can always flip $U_T \to -U_T$ to achieve this, without affecting any of the previous statements we made.

As a small aside, note that not just any collection of Majorana zeromodes remains stable, but only those that satisfy Eq. (4.13). In fact, we see from Eq. (4.10) that $U_T$ has zero trace – so where did all the negative eigenvalues go? We now need to remember that we only looked at the left edge in the above analysis. In fact, the Majoranas located at the right edge must all have a $U_T$ eigenvalue $-1$.[8] If we connect the right edge back to the left edge to restore PBC, this will couple the $2n$ zeromodes, which together now comprise an equal number of $+1$ and $-1$ eigenvectors of $U_T$. Since now tr$M = 0$, it becomes possible to fully gap out the spectrum, as it should: after all, we know that the PBC ground state is unique and gapped.

[8] You can confirm this explicitly by calculating for example $U_T |1, 2N\rangle$ using the definition of $U_T$ in Eq. (4.10).

# 5    Stability to interactions in 1D

We have understood the essential aspects of 1D free fermion topological phases (SPTs) without symmetry (protected by spinless time reversal). In this last section, we study their stability interactions.

We will pursue a rather pedestrian approach, based on checking whether the topological ground state degeneracy in open boundary conditions can be lifted by perturbative interactions. It turns out that this approach gives results that are fully consistent with fancier non-perturbative methods, but we do not prove this.

## 5.1    No symmetry: $\mathbb{Z}_2 \to \mathbb{Z}_2$

Let us recall the Kitaev chain of Sec. 3.3 and following. We briefly review our previous results here, but to keep things interesting we change our notation completely.

The Kitaev chain is a model for a 1D fermion topological phase that only requires locality for its protection (which in turn implies fermion parity symmetry, see the second paragraph of Sec. 2). In periodic boundary conditions (PBC), the ground state is unique and gapped. In open boundary conditions (OBC), we found two Majorana zeromode operators localised to the left (L) and right (R) end of the chain – call them[1] $\gamma_L$ and $\gamma_R$ – that commute with the Hamiltonian. They imply a 2-fold ground state degeneracy: let $|\phi_0\rangle$ be one such ground state in OBC[2]. We saw that we can choose it to be the vacuum of the non-local fermion mode[3]

$$f^\dagger = \frac{1}{2}(\gamma_R - \mathrm{i}\gamma_L),\tag{5.1}$$

meaning that $f|\phi_0\rangle = 0$. Then the other degenerate ground state is

$$|\phi_1\rangle \equiv f^\dagger|\phi_0\rangle.\tag{5.2}$$

We now want to determine if local interactions can open a gap between $|\phi_0\rangle$ and $|\phi_1\rangle$. Let $H$ denote[4] the Kitaev chain Hamiltonian from Eq. (3.15). We add now a small interaction term $O = O^\dagger$ to result in the new Hamiltonian

$$H' = H + O.\tag{5.3}$$

In first-order degenerate perturbation theory, the new ground state energies are obtained by calculting the eigenvalues of the Hermitian matrix

$$\Delta_{mn} = \langle\phi_m|O|\phi_n\rangle,\tag{5.4}$$

where $m, n = 0, 1$. Since $O$ preserves fermion parity symmetry [Eq. (3.6)] by assumption, while $\phi_0$ and $\phi_1$ have different fermion parity by construction[5], this matrix is already diagonal:

[1] In our previous notation, $\gamma_L = \gamma_1$ and $\gamma_R = \gamma_{2N}$.

[2] In our previous notation, $|\phi_0\rangle = |GS_1\rangle$.

[3] In our previous notation, $f^\dagger = c^\dagger_{NL}$.

[4] In our previous notation, $H = H_1$.

[5] The operator $f$ is odd under fermion parity, $PfP^\dagger = -f$.

$$\langle \phi_0 | O | \phi_1 \rangle = \langle \phi_0 | P^\dagger O P | \phi_1 \rangle = - \langle \phi_0 | O | \phi_1 \rangle = 0. \qquad (5.5)$$

We therefore only need to compute the diagonal energy shifts $\Delta E_m \equiv \langle \phi_m | O | \phi_m \rangle$. If they can be made different for the two choices $m = 0, 1$, we have successfully gapped out the degeneracy.

We now show that in fact $E_0 = E_1$ always holds for a local operator $O$. In its most general form, we can write such an operator as

$$O = O_0 \mathbb{1} + \sum_{\alpha\beta} O_{\alpha\beta} \gamma_\alpha \gamma_\beta + \sum_{\alpha\beta\gamma\delta} O_{\alpha\beta\gamma\delta} \gamma_\alpha \gamma_\beta \gamma_\gamma \gamma_\delta + \dots, \qquad (5.6)$$

where the expansion coefficients $O_0, O_{\alpha\beta}, O_{\alpha\beta\gamma\delta}, \dots$, must be chosen such that $O$ is Hermitian[6]. Note that the restriction to free fermions that we had imposed earlier meant that only $O_0$ and $O_{\alpha\beta}$ are allowed to be non-zero, but here we consider a fully interacting $O$ where all higher terms are in principle non-zero as well.

The condition that $O$ be local translates to the requirement that it only involves Majorana operators $\gamma_\alpha$ that are not farther separated than a certain maximum range[7]. Now, if $O$ is an operator acting inside the bulk – meaning it does not involve $\gamma_L$ or $\gamma_R$ – then it commutes with the nonlocal fermion $f$ and therefore

$$\langle \phi_1 | O | \phi_1 \rangle = \langle \phi_0 | f O f^\dagger | \phi_0 \rangle = \langle \phi_0 | O f f^\dagger | \phi_0 \rangle = \langle \phi_0 | O | \phi_0 \rangle, \qquad (5.7)$$

implying the degeneracy is not lifted. On the other hand, if $O$ involves $\gamma_L$ ($\gamma_R$), then it cannot also contain $\gamma_R$ ($\gamma_L$) due to the locality constraint. It is therefore enough to study the situation where $O$ includes $\gamma_L$ but not $\gamma_R$, without loss of generality. We can then sort all terms appearing in the huge sum Eq. (5.6) into two categories: they either *commute* with $\gamma_L = \gamma_1$, or they *anti-commute* with $\gamma_L$.[8] This results in a decomposition

$$O = A + B, \quad [A, \gamma_L] = 0, \quad \{B, \gamma_L\} = 0. \qquad (5.8)$$

Since *both* $A$ and $B$ commute with $\gamma_R$ due to locality, we have

$$\begin{aligned} fA &= \frac{1}{2}(\gamma_R + i\gamma_L)A = A\frac{1}{2}(\gamma_R + i\gamma_L) = Af, \\ fB &= \frac{1}{2}(\gamma_R + i\gamma_L)B = B\frac{1}{2}(\gamma_R - i\gamma_L) = Bf^\dagger. \end{aligned} \qquad (5.9)$$

Since $A$ and $B$ are Hermitian, we also get $f^\dagger A = Af^\dagger$ for free. The last identity we need is

$$f^\dagger B = \frac{1}{2}(\gamma_R - i\gamma_L)B = B\frac{1}{2}(\gamma_R + i\gamma_L) = Bf. \qquad (5.10)$$

We are now ready to evaluate

$$\begin{aligned} \langle \phi_1 | O | \phi_1 \rangle &= \langle \phi_0 | f O f^\dagger | \phi_0 \rangle = \langle \phi_0 | f A f^\dagger | \phi_0 \rangle + \langle \phi_0 | f B f^\dagger | \phi_0 \rangle \\ &= \langle \phi_0 | A | \phi_0 \rangle, \end{aligned} \qquad (5.11)$$

[6] For example, we have seen before that $O_{\alpha\beta}$ must be an imaginary antisymmetric matrix.

[7] For example, $O_{\alpha\beta} = 0$ should hold for all $|\alpha - \beta| > d$ for some given $d$.

[8] For example, $\gamma_2\gamma_3$ commutes with $\gamma_L$, while $\gamma_1\gamma_2\gamma_3\gamma_4$ anti-commutes with $\gamma_L = \gamma_1$.

where we have used $(f^\dagger)^2 = 0$ in the last step. At the same time, we have

$$
\begin{aligned}
\langle \phi_0 | O | \phi_0 \rangle &= \langle \phi_1 | f^\dagger O f | \phi_1 \rangle = \langle \phi_1 | f^\dagger A f | \phi_1 \rangle + \langle \phi_1 | f^\dagger B f | \phi_1 \rangle \\
&= \langle \phi_1 | A | \phi_1 \rangle = \langle \phi_0 | f A f^\dagger | \phi_0 \rangle = \langle \phi_0 | A | \phi_0 \rangle \\
&= \langle \phi_1 | O | \phi_1 \rangle .
\end{aligned}
\tag{5.12}
$$

This shows that no local, parity conserving perturbation $O$ can gap out the two degenerate ground states of the Kitaev chain in OBC, even if it is interacting. Correspondingly, this nontrivial topological phase can still only be deformed into a trivial state – which notably does not have an OBC ground state degeneracy – by closing the gap between the ground states and the excited states.

We conclude that the $\mathbb{Z}_2$ classification that we had found in Sec. 3.2 remains stable to interactions. Formally, we express this as the statement that including interactions maps the classification like this: $\mathbb{Z}_2 \to \mathbb{Z}_2$.

## 5.2 Spinless time-reversal symmetry: $\mathbb{Z} \to \mathbb{Z}_8$

Okay, so we have seen that for a single Kitaev chain – a 1D topological phase that only requires locality – interactions don't do much, at least perturbatively. How about a collection of Kitaev chains? We had argued in Sec. 4 that including spinless time-reversal symmetry stabilises a $\mathbb{Z}$ classification, meaning an integer number $n$ of Kitaev chains are topologically nontrivial, at least at the free fermion level. Does this classification survive when interactions are included?

We will again pursue a perturbative argument, this time starting from a set of $n$ Kitaev chains with associated Majorana zeromodes $\zeta_\lambda$ (located at the left edge) and $\xi_\lambda$ (located at the right edge)[9], $\lambda = 1 \ldots n$. In Sec. 5.1, we had shown that interaction operators that act in the bulk, as well as interactions that involve only a single Majorana zeromode, cannot gap out the ground state degeneracy. The same argument still applies to a stack of $n$ Kitaev chains and so we will not repeat it here. What's new in our present case is that interactions can couple the different Majoranas zeromodes *with each other* locally at a single edge, while all non-interacting couplings were forbidden by time-reversal symmetry. Without loss of generality, we can focus on the left edge where the $\zeta_\lambda$ Majoranas live. All we need to remember is the action of fermion parity [Eq. (3.6)]

$$
P \zeta_\lambda P^\dagger = -\zeta_\lambda,
\tag{5.13}
$$

and time reversal [Eq. (4.6)]

$$
T \zeta_\lambda T^{-1} = \zeta_\lambda,
\tag{5.14}
$$

[9] In our earlier notation from Sec. 4, $\zeta_\lambda = \gamma_1^{(\lambda)}$ and $\xi_\lambda = \gamma_{2N}^{(\lambda)}$.

where we have to remember that $T$ is anti-unitary.

Let's increase $n$ step by step and see what happens.

(1) $n = 1$ is a single Kitaev chain, whose OBC ground state degeneracy is stable – there are no multiple Majorana zeromodes at a single edge that we could couple.

(2) $n = 2$ are two Kitaev chains with two Majorana zeromodes $\zeta_1$ and $\zeta_2$ at the left edge. The only coupling term is

$$O = i\zeta_1\zeta_2, \tag{5.15}$$

which is a non-interacting (= quadratic) term. We have already seen in Sec. 4 that this term is disallowed by time-reversal symmetry. We therefore find that the OBC ground state degeneracy remains stable also in this case.

(3) $n = 3$: Again, the only possible coupling terms are quadratic terms that are all forbidden by time-reversal symmetry. Note in particular that the "interaction"

$$O = i\zeta_1\zeta_2\zeta_3 \tag{5.16}$$

is disallowed as it is nonlocal and anti-commutes with fermion parity symmetry. (Recall from Sec. 3.2 that any local interaction term must preserve fermion parity symmetry = can only contain an *even* number of Majorana operators.) So the degeneracy cannot be lifted.

(4) $n = 4$: Finally things start to get interesting. There seems to be nothing wrong with the interaction term

$$O = \zeta_1\zeta_2\zeta_3\zeta_4. \tag{5.17}$$

However, we have to be careful because at this point our "local" degeneracy at the left edge has grown to $\sqrt{2^4} = 4$, so that the full OBC ground state degeneracy from the two ends of the chain is $2^4 = 16$. Is the operator $O$ above enough to fully gap out this degeneracy?[10] Unfortunately, $O$ is not enough. Define

$$R = i\zeta_1\zeta_2, \quad S = i\zeta_2\zeta_3. \tag{5.18}$$

These operators have eigenvalues $\pm 1$ and satisfy

$$[R, O] = [S, O] = 0, \quad \{R, S\} = 0. \tag{5.19}$$

This algebra immediately implies a remaining ground state degeneracy that we cannot resolve: For example, let $|\Psi\rangle$ be a common eigenstate of $O$ and $R$. Then $S|\Psi\rangle$ has the same energy

[10] There really are no other operators left that we can write down right now that would also be compatible with time reversal and locality.

under $O$, but opposite eigenvalue under $R$, and must therefore be a different yet energetically degenerate eigenstate. We conclude that at least some ground state degeneracy remains: four Kitaev chains are still nontrivial even when we allow for interactions.

(5) $n = 5$: We can generalise the term from Eq. (5.17) to couple 4 Majoranas with each other, leaving out one – this gives 5 possible terms depending on which Majorana we left out[11]. The most general allowed coupling term therefore reads

$$O = a\zeta_1\zeta_2\zeta_3\zeta_4 + b\zeta_2\zeta_3\zeta_4\zeta_5 + c\zeta_3\zeta_4\zeta_5\zeta_1 + d\zeta_4\zeta_5\zeta_1\zeta_2 + e\zeta_5\zeta_1\zeta_2\zeta_3,$$
$$(5.20)$$

where $a \dots e \in \mathbb{R}$ are some coefficients. This interaction Hamiltonian has the form

$$O = A + B + C + D + E, \qquad (5.21)$$

where the matrices $A \dots E$ all *anti-commute* with each other and square to $a^2 \dots e^2$, respectively, times the identity. Therefore, all mixed terms in $O^2$ vanish and we obtain

$$O^2 = a^2 + b^2 + c^2 + d^2 + e^2. \qquad (5.22)$$

But that means that all eigenvalues of $O$ take the form

$$\pm\sqrt{a^2 + b^2 + c^2 + d^2 + e^2}. \qquad (5.23)$$

Since we can choose the parameters $a \dots e$ differently on the left edge and the right edge of the system, this means we get at most 4 different energies – however, overall we have a total of $2^5 = 32$ ground states to contend with, meaning that we are still left with at least an 8-fold degeneracy that cannot be gapped out.

(6) $n = 6$: At this point it becomes cumbersome to enumerate all the possible terms in the perturbation $O$. We clearly need some heavier machinery if we want to keep going. Let's take a step back and think about our approach. We are looking for a perturbation that is local, in the sense that it only acts on the left edge of the system, and manages to fully gap out the ground state degeneracy[12]. For this to happen, the $\sqrt{2^6} = 8$-fold degeneracy *on either edge* must split, independently of what happens on the other edge. In principle, this seems possible as we now have a bunch of interaction terms available.

At this point, however, something surprising happens: it turns out that the restriction to a single edge can alter the symmetry constraints due to fermion parity and time reversal in an essential manner[13]. Let's go through this slowly. Locally, on the left edge,

[11] Again you can convince yourself that there really are no other terms we could write down.

[12] only as long as we also add a similar perturbation to the right edge, of course

[13] The fancy term for this is *symmetry fractionalisation*. The argument is rigorously worked out here:

Lukasz Fidkowski and Alexei Kitaev. Topological phases of fermions in one dimension. *Phys. Rev. B*, 83:075103, Feb 2011. DOI: 10.1103/PhysRevB.83.075103. URL https://link.aps.org/doi/10.1103/PhysRevB.83.075103

we now have 6 Majorana zeromodes and so the *local* fermion parity operator [in analogy to Eq. (3.6)] *on the left edge* (L) reads

$$P_L = (\mathrm{i}\zeta_1\zeta_2)(\mathrm{i}\zeta_3\zeta_4)(\mathrm{i}\zeta_5\zeta_6). \qquad (5.24)$$

You can verify that this definition is compatible with Eq. (5.13). At the same time, time reversal symmetry still satisfies $T\zeta_\lambda T^{-1} = \zeta_\lambda$ as before [Eq. (5.13)] and is anti-unitary. But this means that time-reversal *anti-commutes* with a local fermion parity operation on the left edge:

$$TP_L T^{-1} = -P_L. \qquad (5.25)$$

Correspondingly, assume we have found a unique gapped state of the 6 Majorana fermions on the left edge. It will have some definite local fermion parity = eigenvalue of $P_L$. Applying time-reversal to this state then yields a state that has the opposite local fermion parity, meaning we get at least one other degenerate state. As a consequence, we are not able to completely resolve the degeneracy also in this case.

(7) **$n = 7$**: By now the dimension of the degenerate subspace of ground states is $2^7 = 128$, so we should again look for a general argument rather than just enumerating all possible terms. Let's assume we are able to split the OBC ground state degeneracy by a local interaction term

$$H = O_L + O_R, \qquad (5.26)$$

where $O_L$ only contains Majorana operators on the left edge, and $O_R$ only contains Majorana operators on the right edge. We can again write down the fermion parity operator on the left edge

$$P_L = (\mathrm{i}\zeta_1\zeta_2)(\mathrm{i}\zeta_3\zeta_4)(\mathrm{i}\zeta_5\zeta_6)\zeta_7, \qquad (5.27)$$

where the omission of the factor i for $\zeta_7$ makes the operator Hermitian[14], so that the full fermion parity decomposes as

$$P = \mathrm{i}P_L P_R, \qquad (5.28)$$

where $P_R$ is the similarly defined local fermion parity operator on the right. Note that there is no bulk contribution to $P$, because the bulk is invariant under fermion parity[15]. Now, local fermion parity imposes the constraint

$$[P_L, O_L] = 0, \qquad (5.29)$$

and we also get $[P_L, O_R] = 0$ for free because $O_R$ can only consist of terms with an even number of Majorana operators. At the

[14] At this point, we could just go ahead and make the argument that $P_L$ anti-commutes with $T$, which we already used in the case of $n = 6$ to show that we cannot open a gap. However, we here prefer to avoid $T$ to confirm a more general point: any *odd* number of Kitaev chains cannot be made trivial, even without time-reversal symmetry. This is a direct consequence of the $\mathbb{Z}_2$ classification that remains when time-reversal is broken.

[15] In every of the 7 Kitaev chains, the bulk is constructed from products of the operator $P_j$ in Eq. (3.37); this operator has even fermion parity.

same time, $P_L$ contains an odd number of Majorana operators, and is therefore itself an odd operator under global fermion parity $P$. Correspondingly, if we have found a common eigenstate $|\Psi\rangle$ of $O_L$ and $O_R$ that is a gapped ground state of $H$, then $P_L |\Psi\rangle$ is another eigenstate that is (1) degenerate, because $[P_L, H] = 0$, and (2) orthogonal to $|\Psi\rangle$, because it has opposite *global* fermion parity as measured by $P$. As a consequence, the degeneracy cannot be fully split.[16]

[16] Note that this argument actually works for any odd $n$.

(8) $n = 8$: All our tricks don't work anymore: the local fermion parity operator $P_L$ is now even under time-reversal symmetry, and it is also even under global fermion parity. Might we finally be able to gap the local $\sqrt{2^8} = 16$-fold degeneracy of each edge? Without a smart general argument that forbids this, it makes sense to look for an explicit example that would prove we can open a gap.

You might want to try around for a bit before you continue reading on the next page.

Consider the local operator (acting on the left edge only)

$$\begin{aligned} O &= \zeta_1\zeta_2\zeta_3\zeta_4 + \zeta_5\zeta_6\zeta_7\zeta_8 + \zeta_3\zeta_4\zeta_5\zeta_6 + \zeta_1\zeta_3\zeta_5\zeta_7 \\ &= A + B + C + D. \end{aligned} \tag{5.30}$$

Note that all four operators $A, B, C, D$ commute with each other, square to the identity, and are independent from each other – meaning we cannot express any one as a combination of the others. The eigenstates of $O$ can then be built from the mutual eigenstates of $A, B, C, D$. Since $A, B, C, D$ square to 1, they have eigenvalues $\pm 1$, so that there are $2^4 = 16$ possible eigenvalue combinations. Importantly, this fully exhausts the available Hilbert space. But that must mean that the state where $A, B, C, D$ all have eigenvalue $-1$ is the *unique* ground state of $O$, with eigenvalue $-4$! All other states of the Hilbert space correspond to having at least one of the $A, B, C, D$ eigenvalues come out as $+1$, meaning there is a gap of $1 - (-1) = 2$ between the ground state and the next lowest eigenstate of $O$.

Since we can repeat the same construction also on the right edge, this means our journey has finally come to an end, and we have found a way to completely gap out and thereby trivialise a stack of 8 Kitaev chains without breaking locality or spinless time-reversal symmetry.

In conclusion, interactions reduce the classification of 1D fermion SPT phases with spinless time-reversal symmetry from $\mathbb{Z} \to \mathbb{Z}_8$.

## Acknowledgments

I would like to thank Senthil Todadri and Ken Shiozaki for insightful discussions and remarks. I am also grateful to the organisers of the 2025 Bad Honnef summer school "Symmetry protected topological phases", for which these notes were originally written. This work was supported by a UKRI Future Leaders Fellowship MR/Y017331/1.

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
