# Peer review of "Introduction to some of the simplest topological phases of matter"

_SciPost Physics Lecture Notes_

## Round 1 · Referee Report · Anonymous (Referee 1) · 2025-12-1

Strengths

1) These lecture notes give a very clear and streamlined introduction of free-fermion topological phases and their classification. The material is presented in a sharp, concise, and engaging style. 2) The way the physics is explained requires relatively little technical background, without spending too long on introducing background theory. 3) The particular approach and perspective taken here, which focuses on classification in momentum space, is different from many previous lecture notes and introductory papers on this topic, which typically focus on studying e.g. the SSH model in real space. Different symmetry classes and spatial dimensions are treated in a unified, coherent way.

Weaknesses

1) The scope of these notes is much more limited than the title would suggest. While it is true that free-fermion phases are among the simplest topological phases of matter, based on the title I had expected that some discussion of intrinsic (non-SPT) topological phases, or even non-free-fermionic SPT phases, would be included. The way that the physics is explained (while very much appropriate useful for SPT phases that can be deformed to free-fermions) makes it difficult to describe the more general phenomenology of SPTs, e.g. in spin chains, or to connect to the wider range of topological phases. 2) There is a strong focus on phase classification, without many concrete examples. This makes the discussion somewhat abstract and mathematical at times. 3) There are very few figures, which could be used to diversify the modes of presentation beyond equations+text.

Report

Overall, I am impressed with these lecture notes, as they provide a comprehensive explanation of free-fermion SPTs and their classification. My main concern is point 1) in the "weaknesses" section. While there are several solutions to this, my suggestion would be to just change the title to more accurately describe the scope of what is covered in the notes, rather than adding all the necessary extra material that would be needed to encompass SPTs and topological phases as a whole.

There are a few points in the notes where extra clarity could be provided, either by properly explaining terminology, or by using some simple examples to illustrate the points further, as described in the "Requested changes" section. Most of the other points listed below are minor suggestions of how to make the material even more accessible to readers who are new to this field.

Requested changes

1) Title change (as per "Report" section) 2) Section 1.1 gives a relatively abstract definition of what an SPT phase is. It may be useful to also 'tease' some of their key phenomena, e.g. edge modes, fractionalization of symmetry quantum numbers, gauge anomalies, etc., even if there isn't space to fully explain what all these phenomena are 3) Also in Section 1.1 - a quick definition of what "gapped" means would be useful for readers less familiar with condensed matter terminology 4) In Section 2.2, when introducing the idea of a unit cell, a quick example (and figure?) might help make the construction clearer, e.g. the SSH model? 5) Similarly, at the top of p. 10 when describing a "band structure", a simple example of a band structure might give readers a better sense of what kind of objects are being analysed here. 6) After 2.34, it does not seem immediately clear to a new reader what would happen if $\gamma_1$ and $\gamma_2$ commuted. 7) At the end of Section 2, the result is described as "free fermion SPT phases protected by U(1) symmetry is given by Z in even dimensions, while it is trivial in odd dimensions". Is it really correct to describe Chern insulators as U(1)-protected SPTs? e.g. if one broke U(1) by introducing anomalous $c^\dagger c^\dagger$ terms, would it actually be possible to deform these ground states to a product state, given that the classification in class D is also Z? Some clarification of this point is needed. 8) In Eq (3.5), an explicit representation of $A$ in terms of $\mathcal{H}$ and $\Delta$ would be useful 9) After (3.12), the point that the orthogonal group has disconnected components is glossed over relatively quickly. This seems like an important distinction to make from the classification point of view - that the topology of certain matrix manifolds impacts the classification of free fermion phases (perhaps as a footnote) 10) After 4.7 there is a sentence that ends without being completed

Recommendation

Ask for minor revision

---

## Editorial Decision

awaiting_resubmission